# CENP-E activation by Aurora A and B controls kinetochore fibrous corona disassembly

Susana Eibes [1], Girish Rajendraprasad [1], Claudia Guasch-Boldu [1], Mirela Kubat[1], Yulia Steblyanko[1] & Marin Barisic [1,2] ✉

Accurate chromosome segregation in mitosis depends on multiprotein structures called kinetochores that are built on the centromeric region of sister chromatids and serve to capture mitotic spindle microtubules. In early mitosis, unattached kinetochores expand a crescent-shaped structure called fibrous corona whose function is to facilitate initial kinetochore-microtubule attachments and chromosome transport by microtubules. Subsequently, the fibrous corona must be timely disassembled to prevent segregation errors. Although recent studies provided new insights on the molecular content and mechanism of fibrous corona assembly, it remains unknown what triggers the disassembly of the outermost and dynamic layer of the kinetochore. Here, we show that Aurora A and B kinases phosphorylate CENP-E to release it from an autoinhibited state. At kinetochores, Aurora B phosphorylates CENP-E to prevent its premature removal together with other corona proteins by dynein. At the spindle poles, Aurora A phosphorylates CENP-E to promote chromosome congression and prevent accumulation of corona proteins at the centrosomes, allowing for their intracellular redistribution. Thus, we propose the Aurora A/B-CENP-E axis as a critical element of the long-sought-for mechanism of fibrous corona disassembly that is essential for accurate chromosome segregation.

Faithful chromosome segregation during cell division relies on the ability of chromosomes to bind mitotic spindle microtubules (MTs). This is achieved via kinetochores (KTs), the multiprotein structures that are built on the centromeric region of sister chromatids. In early mitosis, unattached KTs expand a crescent-shaped structure called fibrous corona whose function is to facilitate initial KT-MT attachments[1] and chromosome transport by MTs[2]. Subsequently, the fibrous corona must be timely disassembled to prevent segregation errors[3]. Recent studies revealed that the assembly of fibrous corona relies on the polymerization of ROD-ZW10-Zwilch (RZZ)-Spindly complexes[3–6]. Although these new insights advanced our knowledge on the molecular content and mechanism of fibrous corona assembly, it remains unknown what regulates the disassembly of the outermost and dynamic layer of the KT.

The main mechanism of corona disassembly involves dynein-mediated KT-stripping[7–10], by which corona proteins are transported toward the spindle poles. In addition to reducing KT size, this poleward transport assists in silencing of spindle assembly checkpoint (SAC), thereby promoting chromosome segregation. Two recent studies proposed that RZZ-Spindly[11] and CENP-F-NDE1-NDEL1-LIS1[12] platforms act as "dynein-brakes" that regulate KT-stripping. Earlier studies proposed Aurora B kinase (AurB) to regulate the localization of corona proteins[13–15], however, the exact substrate has not been identified. Kinesin-7/CENP-E is a MT plus-end-directed motor protein required for the congression of peripheral polar chromosomes in early mitosis[16,17]. Moreover, CENP-E is a component of the fibrous corona[18,19] and is phosphorylated by Aurora A (AurA) and AurB in mitosis[20].

[1]Cell Division and Cytoskeleton, Danish Cancer Institute, Copenhagen, Denmark. [2]Department of Cellular and Molecular Medicine, Faculty of Health and Medical Sciences, University of Copenhagen, Copenhagen, Denmark. ✉e-mail: barisic@cancer.dk

Here, we combined site-directed mutagenesis, RNAi, and chemical inhibition with live-cell imaging of fibrous corona stripping and dynamics to address the role of CENP-E phosphorylation as a molecular switch of fibrous corona disassembly. We propose a molecular mechanism of fibrous corona disassembly that relies on AurA/B-CENP-E axis. We show that AurA and AurB phosphorylate CENP-E to release it from an autoinhibited state. At KTs, AurB phosphorylates CENP-E to prevent its premature removal together with other corona proteins by dynein. At the spindle poles, AurA phosphorylates CENP-E to promote chromosome congression and prevent accumulation of corona proteins at the centrosomes, allowing for their intracellular redistribution.

## Results and discussion

### Lack of Aurora A– and B– mediated phosphorylation of CENP-E promotes premature removal of CENP-E from kinetochores and its accumulation at the spindle poles

To monitor CENP-E during mitosis by live-cell imaging, we generated stable U2OS cell lines with doxycycline-inducible expression of GFP-tagged wildtype (WT) CENP-E or a phosphonull CENP-E mutated at the AurA/B-specific phospho-site Threonine 422 (T422A). As previously reported[20], CENP-E-T422A-expressing cells displayed a pseudo-metaphase arrest with several uncongressed chromosomes located at the poles (Fig. 1a, b and Supplementary Movie 1). We analyzed the KT-MT attachment status of these uncongressed chromosomes by immunostaining astrin, a protein that exclusively localizes at KTs attached to MT ends[21]. We detected 85% of KTs to be astrin-negative, indicating that the majority of uncongressed chromosomes in CENP-E-T422A cells are not end-on attached, similar to the results obtained by electron micrographs from cells immunodepleted of CENP-E[22] (Supplementary Fig. 1a, b). Moreover, KTs of polar chromosomes were MAD1-positive, indicating that uncongressed chromosomes in CENP-E T422A-expressing cells prevent progression through mitosis by keeping SAC active (Supplementary Fig. 1a). By measuring the duration of mitosis in nocodazole-treated cells, we observed that CENP-E WT and T422A cells exited mitosis within a similar time frame, implying comparable SAC robustness in both cell lines (Supplementary Fig. 1c).

Interestingly, in addition to congression problems, we observed a strong accumulation of CENP-E-T422A at the spindle poles that started immediately after nuclear envelope breakdown (NEB) (Fig. 1a and Supplementary Movie 1). To address whether this spindle pole-associated accumulation of CENP-E-T422A is coupled with its removal from KTs, we compared the dynamics of GFP-CENP-E-WT and -T422A intensities at KTs. For this purpose, we incorporated a stable expression of CENP-A-mCherry as a non-dynamic KT marker to the above-described GFP-CENP-E cell lines and performed spinning-disk confocal live-cell imaging (Fig. 1c and Supplementary Fig. 1d and Supplementary Data 1). Remarkably, CENP-E-T422A was removed from KTs three times faster than CENP-E-WT (half-life at KTs = 5.4 ± 0.6 min for T422A, compared to 15.2 ± 1.6 min for WT). Moreover, whereas a smaller fraction of CENP-E-WT remained at congressed KTs, this pool was drastically reduced in CENP-E-T422A cells, indicating that AurA/B-mediated phosphorylation is essential for maintaining CENP-E at KTs (Fig. 1c and Supplementary Fig. 1d).

Given that both AurA and AurB can phosphorylate CENP-E at T422[20], we used small molecule inhibitors to unveil the individual roles of these kinases in CENP-E regulation. Since AurA is essential for bipolar spindle assembly[23], we monitored the impact of kinase inhibition on CENP-E in monopolar spindles, induced by inhibition of kinesin-5/EG5 using S-trityl-L-cysteine (STLC). Close to 60% (57.67 ± 2.18 %) of the cells treated with the AurA inhibitor MLN8054 showed a transient polar accumulation of CENP-E, whereas this effect was not detected in any of control cells (Fig. 1d, e and Supplementary Movie 2), suggesting that AurA regulates CENP-E association with spindle poles. Next, we analyzed the effect of AurB inhibition on CENP-E localization at KTs, which was previously shown to partially depend on AurB

activity[15]. To distinguish the impact of AurB on CENP-E loading to or removal from KTs, we compared KT-based localization of CENP-E in nocodazole-treated cells that lack MTs, with STLC-induced monopoles that form KT-MT attachments. After the addition of AurB inhibitor ZM447439, the immunofluorescence-based CENP-E intensity in nocodazole-treated cells was slightly but statistically non-significantly reduced, suggesting that AurB activity is not essential for CENP-E loading to KTs and/or expansion of fibrous corona (Fig. 1f, g and Supplementary Data 1). In striking contrast to nocodazole-treated cells, AurB inhibition in monopoles reduced CENP-E intensity at KTs by 70% (Fig. 1f, g, Supplementary Fig. 1e, and Supplementary Data 1). These data reveal a major role of AurB in prevention of premature removal of CENP-E from KTs, which is in line with earlier work showing a stronger effect of AurB inhibition on KT-bound CENP-E in the presence of MTs[15].

Altogether, these data suggest that AurB holds CENP-E at KTs in early mitosis, while AurA prevents the accumulation of CENP-E at the spindle poles once it arrives there (Fig. 1h). Therefore, the remaining KT-based activity of AurB may be the reason why we observed only a transient polar accumulation of CENP-E upon AurA inhibition. Under these conditions, AurB activity at KTs close to the poles may compensate for the lack of AurA, thereby releasing initially accumulated CENP-E from the spindle poles. Indeed, when we co-inhibited AurA and AurB kinases, CENP-E was absent from KTs, but accumulated at the pole, mimicking the behavior of CENP-E-T422A (Supplementary Fig. 1f).

### Removal of phospho-null CENP-E from kinetochores and its accumulation at the spindle poles require microtubules

To examine the premature removal of CENP-E-T422A from KTs in more detail, we performed high temporal resolution imaging. From the beginning of prometaphase, we observed comets of non-phosphorylated CENP-E being transported poleward from KTs and then accumulating at the poles (Fig. 2a and Supplementary Movie 3). Instead of a continuous streaming from KTs, CENP-E-T422A traveled in bigger chunks, as previously observed with MAD2 and ZW10 stripping[24,25], suggesting that fibrous corona can be disassembled and transported in bigger fragments. This result, together with different effects of AurB on STLC- versus nocodazole-treated cells, suggests that the observed premature removal of CENP-E-T422A depends on MTs. Moreover, the CENP-E-T422A comets traveled with a velocity of 160 ± 26 nm/s (Fig. 2b, Supplementary Data 1, and Supplementary Movie 3), which matches the velocity of in-vitro reconstituted human cargo-bound dynein[26].

To test whether the poleward transport of non-phosphorylated CENP-E is MT-dependent, we performed live-cell imaging of CENP-E-WT and CENP-E-T422A cells treated with nocodazole before entering mitosis. In the absence of MTs, CENP-E-WT and CENP-E-T422A were equally recruited at KTs upon NEB, while no spindle pole accumulation of CENP-E-T422A was detected, confirming that the observed removal of CENP-E-T422A is MT-dependent (Fig. 2c and Supplementary Movie 4). Furthermore, we tested whether CENP-E-T422A can be reloaded on previously bi-oriented KTs. To do so, we used nocodazole to depolymerize MTs in pseudo-metaphase-arrested CENP-E-T422A cells. 10 min after the addition of drug, CENP-E-T422A was detected at KTs, which was coupled to shrinkage of the spindle, indicating effective MT depolymerization (Fig. 2d, e and Supplementary Data 1). Thus, although CENP-E-T422A is recruited to KTs in the absence of MTs, its removal is MT-dependent.

Corona expansion is driven by conformational changes in Spindly, which initiates RZZ-Spindly oligomerization[3–5]. This depends on MPS1 kinase activity but is independent of AurB or Plk1-mediated phosphorylation[3,6]. To test whether the faster removal of CENP-E-T422A from KTs was merely a consequence of its impact on fibrous corona assembly, we analyzed fibrous corona formation. We observed

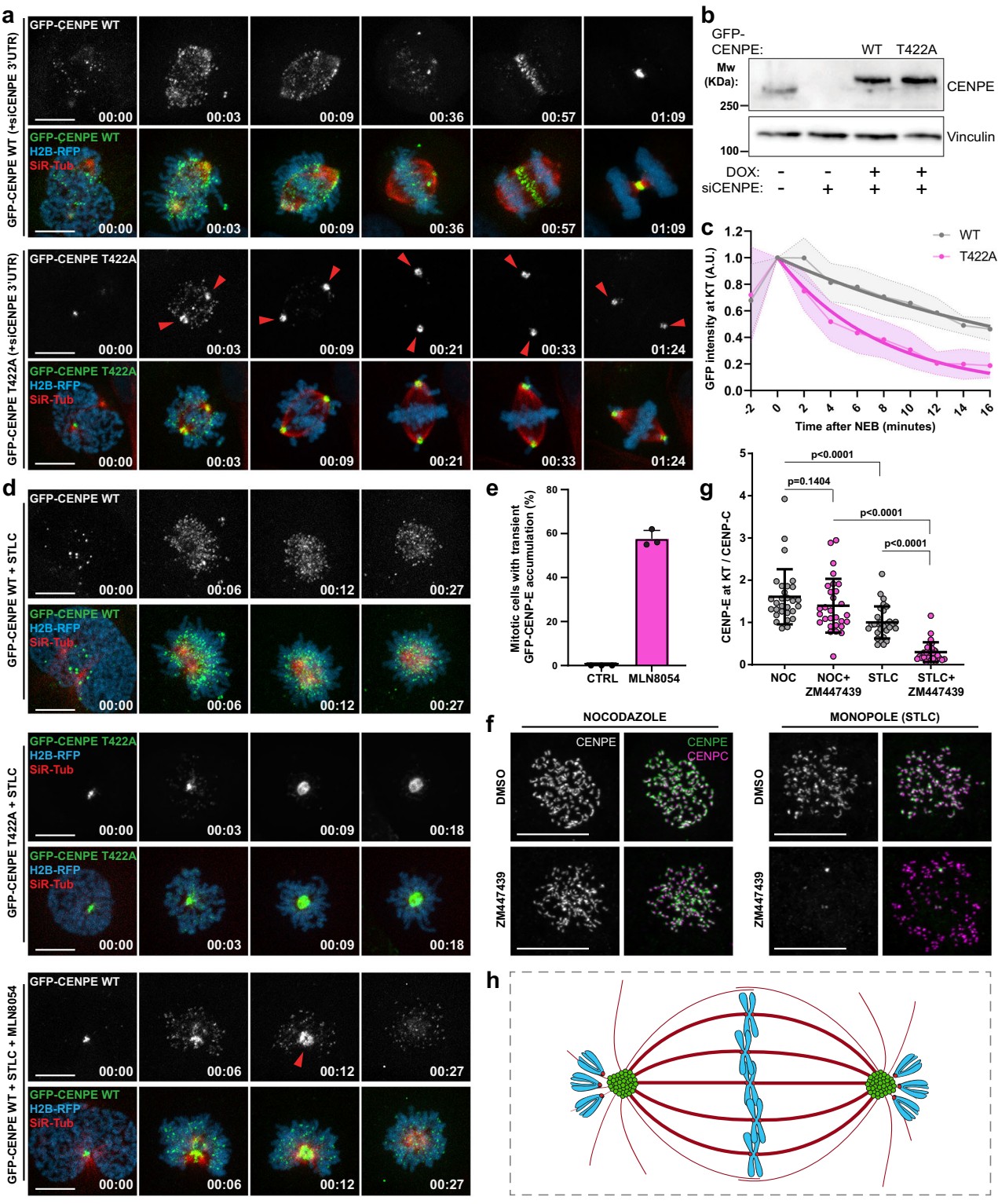

equally efficient assembly and expansion of corona in CENP-E-WT and CENP-E-T422A cells (Supplementary Fig. 2a, b). In addition, we tested corona assembly in HeLa cells, which similar to RPE-1 cells can expand their fibrous coronas beyond the crescent-shaped structure observed in U2OS cells, forming full rings around centromeres. We transiently transfected HeLa cells with GFP-CENP-E-WT or GFP-CENP-E-T422A in HeLa cells depleted of endogenous CENP-E and added nocodazole to allow corona expansion. Both CENP-E intensity at KT (0.97 ± 0.77 in WT; 0.96 ± 0.57 in T422A) and corona volume (0.093 ± 0.054 μm³ in WT; 0.083 ± 0.051 μm³ in T422A) showed no difference between the

two conditions, indicating that corona assembly does not depend on CENP-E phosphorylation (Fig. 2f, g and Supplementary Fig. 2a, b, and Supplementary Data 1).

We next examined the localization of phosphorylated CENP-E (pT422) within the corona using the phospho-antibody against CENP-E T422[20] (Fig. 2h-j and Supplementary Fig. 2c). Performing immuno-fluorescence in nocodazole-treated cells with expanded corona, we revealed that pT422 decorated only the inner part of corona, adjacent to the centromeric region where AurB resides (Fig. 2h-j and Supplementary Fig. 2c). This suggests that the impact of CENP-E phosphorylation on its

**Fig. 1 | Lack of Aurora A- and B- mediated phosphorylation of CENP-E promotes premature removal of CENP-E from kinetochores and its accumulation at the spindle poles. a** Representative spinning-disk confocal time-series of mitosis in U2OS GFP-CENP-E WT/T422A cells. H2B-RFP and SiR-tubulin are used to visualize DNA and MTs, respectively. Red arrowheads indicate spindle pole accumulation of CENP-E mutant. Time: hour:min. Scale bar: 10 μm. **b** Immunoblot of inducible expression of GFP-CENP-E along with depletion of endogenous CENP-E. **c** GFP-CENP-E intensity at KTs over time after NEB in U2OS cells expressing CENP-A-mCherry. Dots and dashed lines represent mean values and SD respectively. The solid thick line represents the fitted curve. N (number of cells, number of independent experiments): GFP-CENP-E WT (10, 4), GFP-CENP-E T422A (10, 4). **d** Representative time-series of GFP-CENP-E WT/T422A localization in monopolar spindles. Time scale: hour:min. Bottom panel shows GFP-CENP-E WT distribution in Aurora A-inhibited cells. Red arrowhead emphasizes transient spindle pole accumulation of CENP-E WT in the absence of Aurora A activity. Time: hour:min. Scale bars: 10 μm. **e** Quantification of the percentage of mitotic cells with transient

accumulation of GFP-CENP-E WT at the spindle poles. The values are plotted with mean and SEM. N (number of cells, number of independent experiments): CTRL (32, 3) MLN8054 (28, 3). **f** Maximum intensity projected confocal images of endogenous CENP-E at KTs in U2OS cells undergoing the indicated treatments. Scale bars: 10 μm. **g** Quantification of endogenous CENP-E intensity normalized to CENP-C intensity at KTs. Violin plots with median (thick dashed lines) and quartiles (light dashed lines) represent the average KT intensities in cells. N (number of KTs, number of cells, number of independent experiments): NOC (1036, 29, 3), NOC + ZM447439 (1010, 29, 3), STLC (1044, 29, 3), STLC + ZM447439 (1063, 29, 3). Statistical significance was determined by the Mann–Whitney U-test (unpaired, two-tailed; no normal distribution). p values are indicated. **h** Illustrated model of the impact of T422A mutation on CENP-E localization and chromosome congression. Non-phosphorylatable CENP-E (green) is prematurely removed from KTs, and it accumulates at the spindle poles. Cells are arrested in a pseudo-metaphase with uncongressed polar chromosomes.

removal from KTs is spatially regulated by AurB, depending on the distance between CENP-E molecules and AurB, as proposed for the general mode of action of AurB[27,28].

## Lack of CENP-E phosphorylation at T422 promotes dynein-mediated stripping of CENP-E from kinetochores

Altogether, these results demonstrate that non-phosphorylated CENP-E is prematurely removed from KTs in a MT-dependent manner, with the velocity of its poleward transport corresponding to the velocity of dynein. To further examine whether non-phosphorylated CENP-E is removed poleward by dynein, we used RNAi to deplete Spindly, a dynein adaptor protein that is required to recruit dynein to KTs[25,29–32]. As expected, Spindly depletion caused chromosome congression problems[25,31] and delayed the removal of CENP-E-WT from KTs (Supplementary Fig. 3a). Importantly, Spindly knockdown drastically increased the lifetime of CENP-E-T422A at KTs, delaying its removal from KTs, (Fig. 3a–c, Supplementary Fig. 3b, and Supplementary Data 1 and Supplementary Movie 5) and preventing its subsequent accumulation at the spindle poles (Fig. 3a, d and Supplementary Fig. 3b, c and Supplementary Data 1 and Supplementary Movie 5). To confirm that the effect of Spindly depletion was a consequence of impeding the dynein recruitment, we performed a rescue experiment in Spindly-depleted CENP-E-T422A cells by expressing RNAi-resistant FLAG-Spindly-WT or FLAG-Spindly-F258A mutant. Spindly-F258A contains a point mutation in the Spindly box (SB) domain that weakens its binding to dynein and reduces dynein loading to KTs[31]. Whereas Spindly-WT successfully recovered spindle pole accumulation of CENP-E-T422A in most cells (98 ± 2%), the recovery achieved by the SB mutant was less efficient (33 %), confirming that the poleward transport and polar accumulation of CENP-E-T422A is driven by KT-associated dynein (Fig. 3e, Supplementary Fig. 3d, and Supplementary Data 1). Although our data indicate that CENP-E and dynein form a complex in mitosis, such interaction has never been shown. Therefore, we performed a co-immunoprecipitation experiment in nocodazole-arrested, mitotically enriched cell extracts, which revealed that GFP-CENP-E co-immunoprecipitates with dynein/dynactin complex components, such as dynein light intermediate chain 1 (LIC1) and dynactin subunit p150/Glued. Thus, this set of experiments indicates that dynein and CENP-E form a complex during mitosis that promotes CENP-E stripping from KTs (Fig. 3f).

## Lack of CENP-E phosphorylation initiates dynein-mediated stripping of corona

Next, we examined whether CENP-E in general regulates KT-bound dynein, either via its loading or removal. For this purpose, we generated HeLa cells stably co-expressing dynein heavy chain (DHC)-GFP and CENP-A-mCherry, which allowed us to quantify the dynamics of KT-bound dynein in control and CENP-E-depleted living cells. Although

under both conditions dynein was successfully recruited at KTs immediately upon NEB, CENP-E-depleted cells displayed a much quicker removal of DHC-GFP from KTs (half-life at KTs: 0.8 ± 0.08 min in siCENP-E, compared to 4.13 ± 0.74 min in controls), demonstrating that CENP-E is dispensable for dynein loading, but required for its maintenance at KTs (Fig. 4a, b, Supplementary Data 1, and Supplementary Movie 6).

We next asked whether the stripping of other corona proteins is regulated by CENP-E phosphorylation. For optimal observation of KT-stripping, we used STLC to induce monopolar spindles in CENP-E-WT and CENP-E-T422A cells. Monopolar cells form KT-MT attachments with approximately two thirds of KTs being either laterally or end-on attached[21], which allows visualization of corona stripping. Moreover, the radial distribution of chromosomes facilitates observation of eventual polar accumulation, which could otherwise be obstructed by uncongressed polar chromosomes in bipolar spindles. In addition to the GFP-CENP-E signal, we examined the intensity and localization of several KT proteins, including corona components dynein/dynactin (p150), Spindly, RZZ (ZW10) and MAD1, as well as KT- but non-corona protein BUBR1, which recruits CENP-E to the outer KT[33–35]. CENP-E-WT, together with other studied KT proteins displayed normal localization at KTs (Fig. 4c and Supplementary Fig. 4a, b). Importantly, all corona proteins accumulated at the poles together with CENP-E-T422A, whereas BUBR1 remained at KTs (Fig. 4c–e, Supplementary Fig. 4c, and Supplementary Data 1). As expected, and similar to CENP-E-T422A, the corona proteins and BUBR1 remained at KTs in nocodazole-treated cells that lack MTs (Supplementary Fig. 4b). Thus, these experiments demonstrate that the absence of CENP-E phosphorylation by AurA/B promotes corona stripping. Upon the disassembly of corona, this phosphorylation is required to prevent polar accumulation of corona proteins, allowing their intracellular redistribution.

## Aurora A and B kinases activate CENP-E by releasing it from an auto-inhibited state

Although we uncovered the functional relevance of AurA/B-mediated CENP-E phosphorylation in the context of fibrous corona disassembly, little is known about the impact of this phosphorylation on intra- and inter-molecular properties of CENP-E. Initially, AurA/B phosphorylation of CENP-E was shown to prevent the binding of the protein phosphatase 1 (PP1) to the CENP-E residues overlapping with this phosphorylation site[20]. In agreement with our data, this phosphorylation was shown to be essential for CENP-E function, as non-phosphorylated CENP-E failed to congress polar chromosomes to the metaphase plate, even though its recruitment at KTs was not altered[20]. Using in-vitro reconstitution of *Xenopus* CENP-E, another study proposed that CENP-E can undergo an autoinhibited conformational state in which the C-terminal tail interacts with and blocks the N-terminal motor domain[36], as it occurs in other kinesins, like

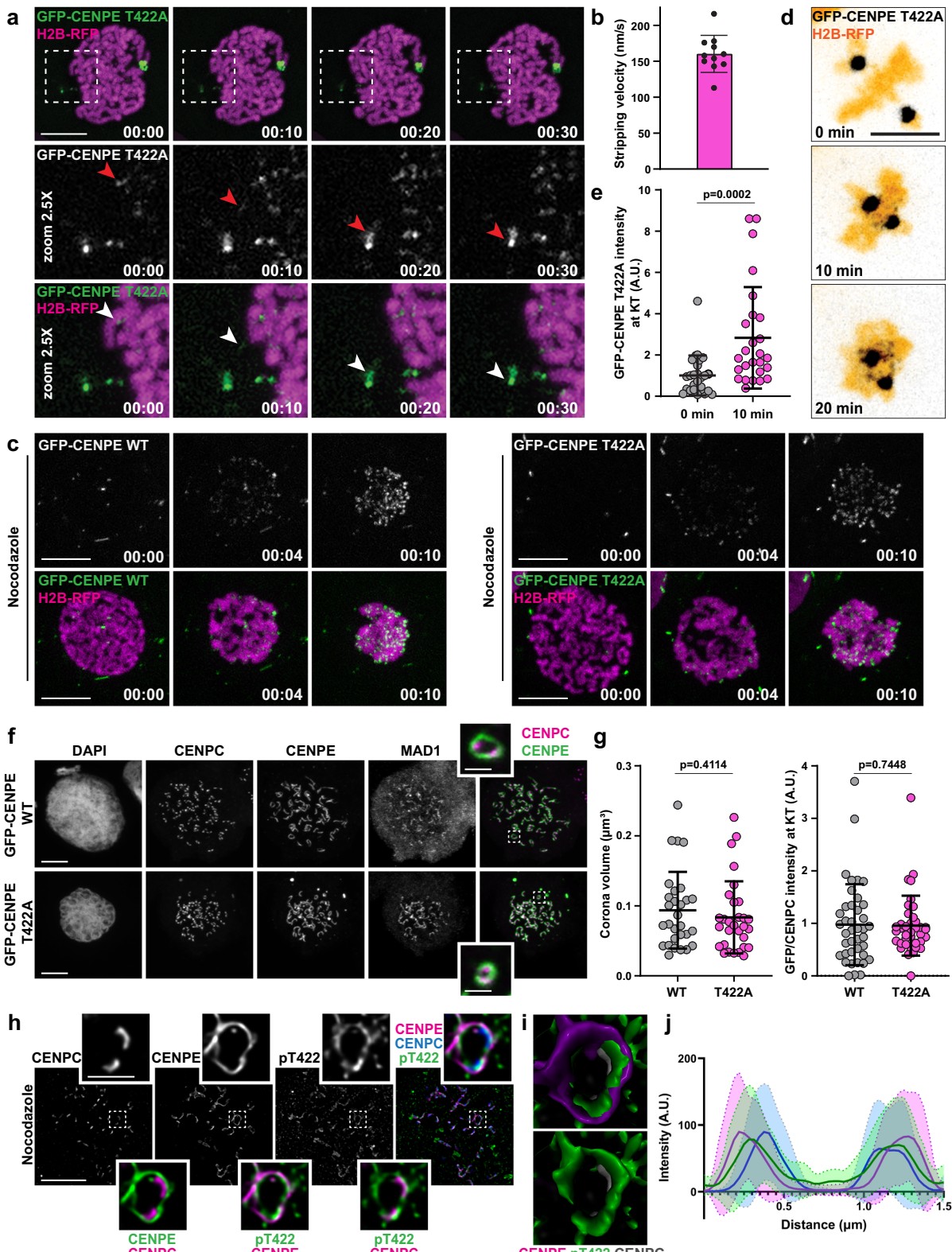

kinesin-1[37]. In-vitro, this autoinhibition can be relieved by MPS1 and CDK1 phosphorylation of the CENP-E C-terminal tail[36]. Moreover, it has been proposed that shortening the long coiled-coil stalk of CENP-E promotes its head-to-tail interaction, autoinhibiting the motor activity[38]. Consistent with the autoinhibition model, electron micrographs of *Xenopus* CENP-E revealed a small percentage of the molecules to be present in a folded state, in which the C-terminal tail

appeared to bind the coiled-coil region adjacent to the motor domain corresponding to the location of the AurA/B phospho-site[38]. Finally, similar to the behavior of CENP-E-T422A that we present here, several studies showed that chemically inhibited CENP-E accumulated at the poles[38,39].

Taken together, these previous data led us to hypothesize that CENP-E-T422A is autoinhibited. The lack of CENP-E phosphorylation at

**Fig. 2 | Removal of phospho-null CENP-E from kinetochores and its accumulation at the spindle poles require microtubules. a** High temporal resolution spinning-disk confocal time-series of GFP-CENP-E T422A at KTs in U2OS cells. Time: min:sec. Scale bar: 10 µm. Red and white arrowheads indicate a CENP-E T422A comet. **b** Quantification of GFP-CENP-E T422A stripping velocity represented with mean ± SD. A total of 11 comets in 9 cells from 3 independent experiments were analyzed. **c** Representative time-series of U2OS GFP-CENP-E WT/T422A cells entering mitosis in the absence of MTs. Time: hour:min. Scale bar: 10 µm. **d** Time-lapse images of GFP-CENP-E T422A reloading at KTs after MT depolymerization. Time: hour:min Scale bars: 10 µm. **e** Quantification of GFP-CENP-E T422A at the KTs over time after the addition of nocodazole. Average KT intensities in cells are represented with mean ± SD. *N* (number of KTs, number of cells, number of independent experiments): $t = 0$ min (520, 26, 3), $t = 10$ min (525, 226, 3). Statistical significance was determined by the Mann–Whitney *U*-test (unpaired, two-tailed; no normal distribution). *p* values are indicated. **f** Representative maximum intensity-

projected confocal images of expanded fibrous coronas in HeLa cells transiently transfected with GFP-CENP-E WT/T422A. Insets display single KT pairs with expanded coronas. Scale bar: 5 µm; scale bar in insets: 1 µm. **g** Quantification of corona volume and GFP intensity normalized to CENP-C intensity at KTs. Average KT intensities in cells are represented with mean ± SD. N (number of coronas, number of cells, number of independent experiments) · Corona volume: WT (1038, 29, 3), T422A (1063, 30, 3); KT intensity: WT (899, 40, 3), T422A (781, 38, 3). Statistical significance was determined by the Mann–Whitney *U*-test (unpaired, two-tailed; no normal distribution). **h** Structured illumination microscopy (SIM) images of phosphorylated CENP-E at expanded coronas of RPE cells, detected using the anti-phospho-T422 antibody. Scale bar: 5 µm; scale bar in insets: 1 µm. **i** 3D surface rendering model of the fibrous corona represented in **h. j** Intensity profile plot of 40 coronas from 10 different cells from 2 independent experiments as represented in **h**. Thick and dashed lines represent averages and SD, respectively.

T422 could lead to autoinhibition via its direct effect on CENP-E conformation, and/or by promoting the binding of PP1[20], which could then dephosphorylate the C-terminus of CENP-E[36]. To discriminate between these two possibilities, we first used site-directed mutagenesis to test the role of the Tryptophan 423 (W423) of CENP-E, since mutating the corresponding residue in *Xenopus* CENP-E (W425) strongly reduced the binding of PP1[20]. If CENP-E-T422A promoted the autoinhibition via PP1 binding, CENP-E-T422A/W423A is expected to rescue the phenotype observed upon the expression of CENP-E-T422A. Live-cell imaging of U2OS cells expressing a single point mutant GFP-CENP-E-W423A displayed localization and behavior similar to GFP-CENPE-WT (Supplementary Fig. 5a, b). On the other hand, a double point mutant GFP-CENP-E-T422A/W423A-expressing cells arrested in mitosis, displaying chromosome congression problems and polar accumulation of GFP-CENPE-T422A/W423A, thus failing to rescue the phenotype induced by GFP-CENPE-T422A expression (Supplementary Fig. 5a, b). This suggests that the absence of T422 phosphorylation does not inhibit CENP-E by promoting the binding of PP1.

To test whether the lack of T422 phosphorylation inhibits CENP-E via a direct effect on intramolecular properties of CENP-E, we generated two constructs covering the N-terminal region of CENP-E (amino acids 1-859), with and without the T422A mutation, and the C-terminal region of CENP-E (amino acids 1799-2701) (Fig. 5a). To ensure the phosphorylation of CENP-E N-terminal domain, we used mitotically enriched, nocodazole-arrested HEK293 cells and immunoprecipitated the C-terminal domain. In the absence of T422 phosphorylation, the N-terminal domain co-immunoprecipitated stronger with the C-terminal domain, indicating that non-phosphorylated CENP-E has a higher predisposition for intra-molecular interaction between its N- and C-termini that may promote autoinhibition (Fig. 5b). Moreover, CENP-E (1799-2701) showed a dominant negative behavior that mimicked CENP-E-T422A phenotype, displaying reduced KT localization of endogenous CENP-E, followed by CENP-E accumulation at the spindle poles and chromosome congression problems (Supplementary Fig. 6a and Supplementary Movie 7). This suggests that the C-terminal fragment interacts and inhibits endogenous CENP-E, which is in line with a dominant negative effect caused by the expression of different C-terminal KT-targeting domain constructs of CENP-E[34]. An alternative explanation is that, because it contains the KT-targeting domain, the C-terminal fragment induces its dominant-negative effects by displacing the endogenous CENP-E from KTs. However, the latter model could not explain polar accumulation of CENP-E.

If our model, in which non-phosphorylated CENP-E is essentially an inactive motor, is correct, the above-described effects of CENP-E-T422A should be phenocopied by chemical inhibition of CENP-E. To address this, we used two CENP-E inhibitors with different modes of action: 1) GSK923295, which induces CENP-E to be bound to MTs in a rigor-like state[40], and 2) Cmpd-A, which acts as an ATP competitor that inhibits the ATPase activity of CENP-E, without increasing its binding to

MTs[41]. To quantify the effect of CENP-E inhibitors, we performed immunofluorescence in monopoles of control cells and cells where dynein-mediated stripping was obstructed by Spindly RNAi. We quantified the ratio between the CENP-E intensity at spindle pole and total CENP-E intensity in monopoles (Supplementary Fig. 4c). The intensity ratio in GSK923295- and Cmpd-A- treated cells was comparable to the results from CENP-E-T422A cells (Fig. 5c, d and Supplementary Data 1), demonstrating strong accumulation of CENP-E at the poles. Moreover, no additive effect was detected when either inhibitor was added to CENP-E-T422A cells, further suggesting that CENP-E-T422A is autoinhibited.

To confirm the effect of CENP-E inhibitors on CENP-E localization, we utilized live-cell imaging of GFP-CENP-E-WT cells entering mitosis in the presence of CENP-E inhibitiors. As observed in immuno-fluorescence images of monopoles, both GSK923295 and Cmpd-A promoted accumulation of CENP-E-WT at the poles of bipolar living cells (Fig. 5e and Supplementary Movie 8).

Intriguingly, whereas Cmpd-A-mediated polar accumulation of CENP-E depended on dynein-driven KT-stripping, GSK923295 promoted CENP-E accumulation at the spindle poles even in the absence of KT-bound dynein (Fig. 5c, d and Supplementary Fig. 6b and Supplementary Data 1 and Supplementary Movie 9). Given that GSK923295 locks CENP-E on MTs in a rigor-like bound state, we hypothesized that GSK923295-treated cells, in addition to dynein-driven KT-stripping, accumulate CENP-E at the poles via MT poleward flux[42]. To address that, we used low-dose nocodazole to effectively reduce MT-flux[43] in metaphase-arrested cells (MT-flux: $0.18 \pm 0.09$ µm/min in 20 nM nocodazole, compared to $0.62 \pm 0.17$ µm/min in controls) (Supplementary Fig. 6c). Whereas reduced MT-flux had no effect on polar accumulation of CENP-E induced by Cmpd-A, it disrupted GSK923295-induced spindle pole accumulation, keeping CENP-E bound along the spindle MTs (Supplementary Fig. 6d, e). These data demonstrated that, unless it is rigor-bound to MTs (GSK923295), the autoinhibited (CENP-E-T422A) or chemically inhibited CENP-E (Cmpd-A) is transported from KTs to the spindle poles by dynein, where it accumulates, as it requires motor activity to leave the poles (Supplementary Fig. 6f, g). In contrast to CENP-E-WT, the polar accumulation of CENP-E-T422A was prevented in the absence of KT-bound dynein in GSK923295-treated cells (Fig. 5c, d, Supplementary Fig. 6b, and Supplementary Movie 9), suggesting that due to its conformational change, the autoinhibited, folded CENP-E cannot bind MTs in a rigor-like state upon GSK923295 treatment, and therefore cannot be transported to the poles via MT-flux.

Taken together, our co-immunoprecipitation and chemical inhibition-based results indicate that AurA/B promote motor activation of CENP-E by releasing it from an autoinhibited conformational state. Similar to CENP-E depletion and inhibition[17,40,44], non-phosphorylated, autoinhibited CENP-E causes failure in congression of polar chromosomes.

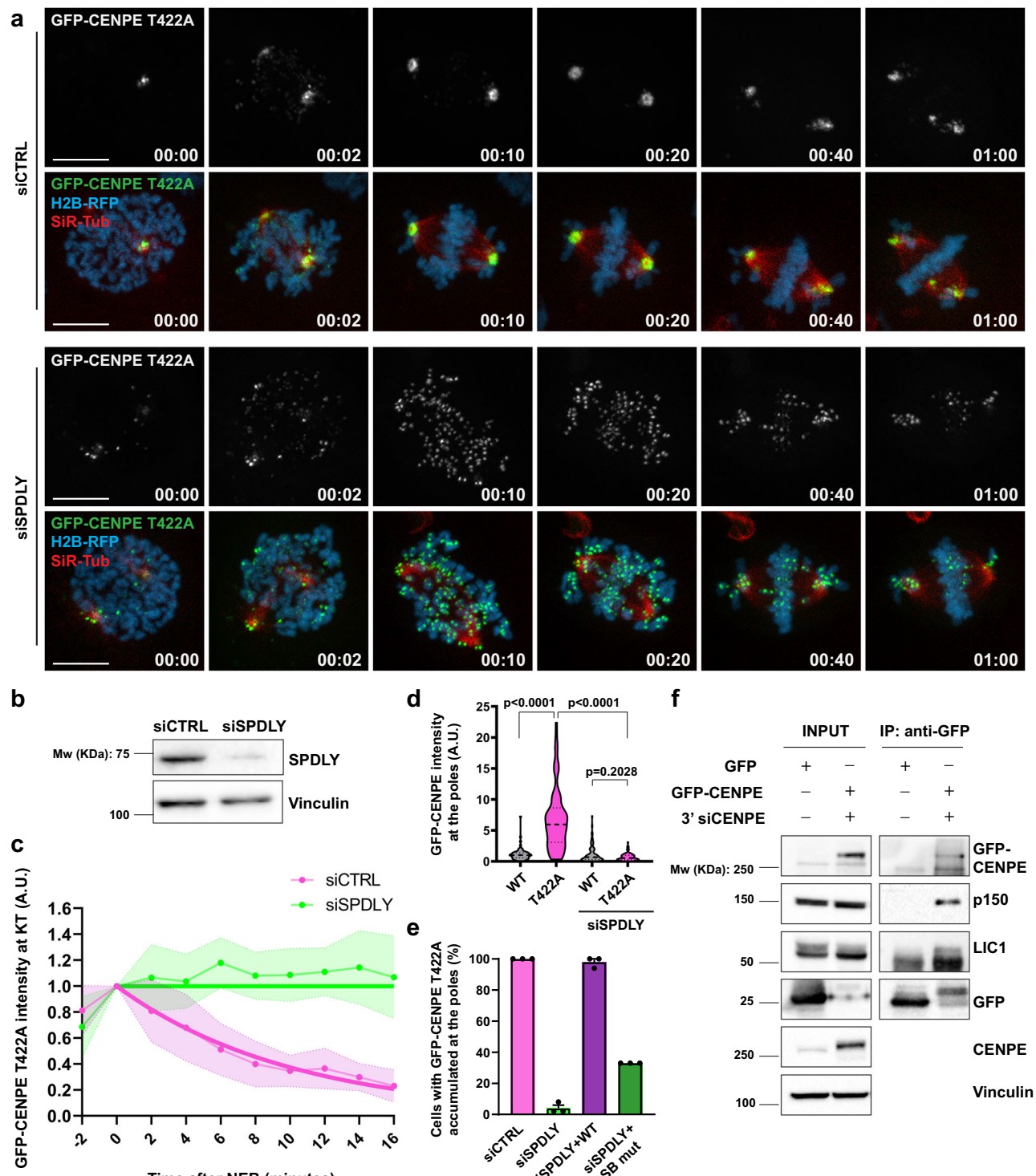

**Fig. 3 | Lack of CENP-E phosphorylation at T422 promotes dynein-mediated stripping of CENP-E from kinetochores. a** Representative spinning-disk confocal time-series of control and Spindly-depleted U2OS cells expressing GFP-CENP-E T422A. Time scale: hour:min. Scale bar: 10 μm. **b** Immunoblot of Spindly depletion in GFP-CENP-E T422A cells. Four independent experiments showed similar results. **c** GFP-CENP-E T422A intensity at KTs at different time points after NEB in CENPA-mCherry-expressing cells. Dots and dashed lines represent mean values and SD, respectively. The solid thick line represents the fitted curve. *N* (number of cells, number of independent experiments): T422A + siCTRL (10, 4), T422A + siSPDLY (10, 2). **d** Immunofluorescence-based quantification of GFP-CENP-E intensities at the spindle poles in metaphase/pseudo-metaphase. Violin plots with median (thick dashed lines) and quartiles (light dashed lines) are presented. *N* (number of spindle poles, number of independent experiments) WT (66, 3) T422A (84, 3), WT + siSPDLY (90, 3), T422A + siSPDLY (79, 3). Statistical significance was determined by the Mann–Whitney *U*-test (unpaired, two-tailed; no normal distribution). *p* values are indicated. **e** Percentage of pseudo-metaphases with GFP-CENP-E T422A accumulated at the pole, represented with mean and SEM. *N* (number of cells, number of independent experiments): siCTRL (150, 3), siSPDLY (150, 3), siSPDLY + SPDLY WT (32, 3) and siSPDLY + SPDLY SB mutant (30, 3) in 3 immunostainings. Scale bar: 10 μm. **f** Co-immunoprecipitation of GFP-CEN-PE WT with two subunits of the dynein complex dynactin/p150 and dynein light intermediate chain/LIC1 from mitotic enriched HEK293 cells. Three independent experiments showed similar results.

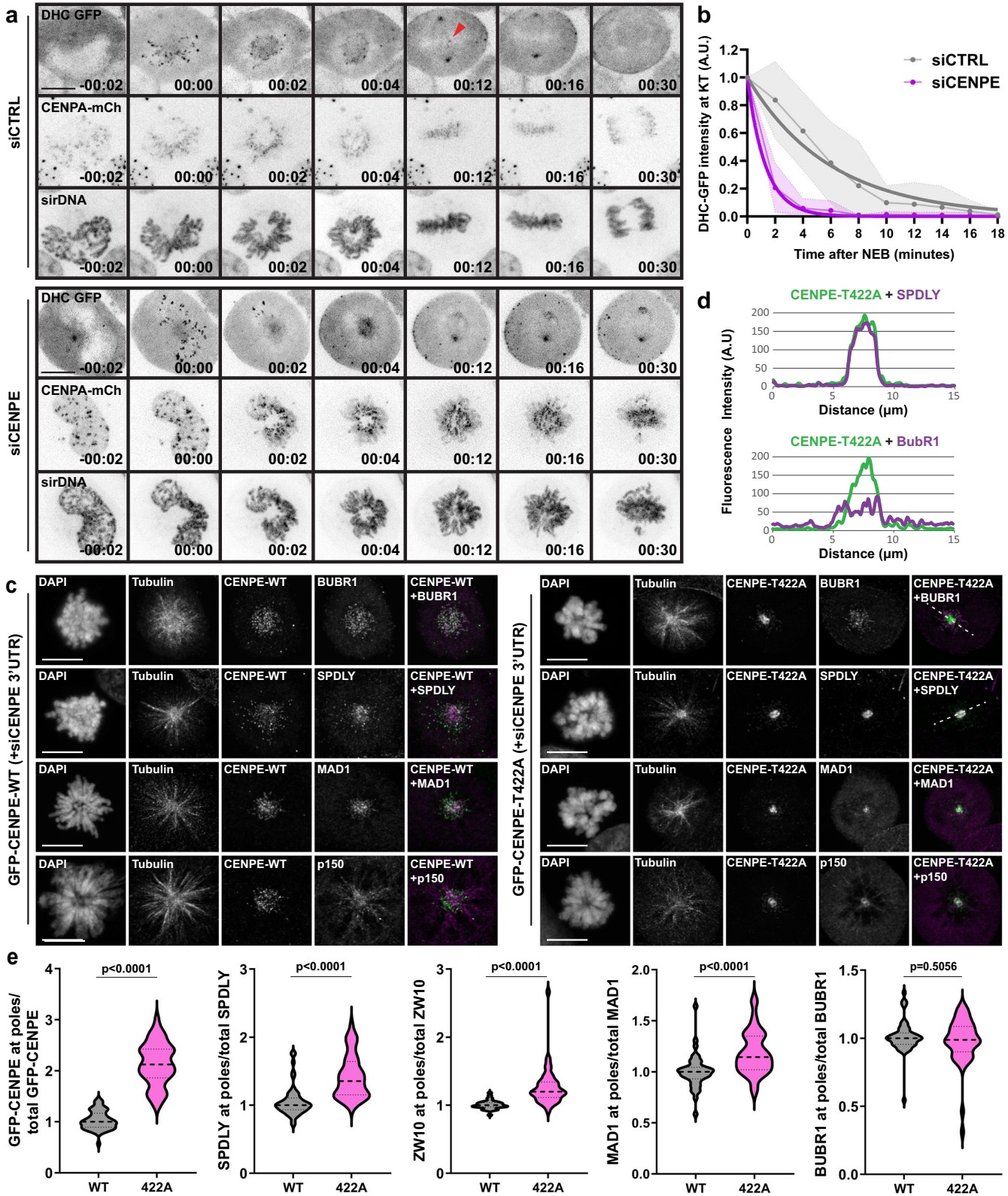

### The interplay between the corona disassembly and chromosome poleward movement

Collectively, our results suggest that lateral KT-MT attachments are sufficient to initiate corona stripping when CENP-E is inactivated, which is in line with a recent study showing that the dynein-mediated removal of corona proteins does not require stable end-on KT-MT attachments[45]. Of note, even though CENP-E inactivation promotes corona stripping by dynein, we still observe poleward movement of

chromosomes in CENP-E T422A or CENP-E-inhibited cells. There are few possibilities that could explain how chromosomes move poleward regardless of the initiation of corona stripping and associated dynein removal. First, removal of corona proteins could release lateral KT-MT attachments, leading to detached KTs that reload the corona components (Fig. 2d, e) and therefore reinitiate lateral attachment. This may be sufficient to promote a stepwise poleward movement. Second, the remaining chromosome poleward movement could be driven

**Fig. 4 | Lack of CENP-E phosphorylation initiates dynein-mediated stripping of corona. a** Spinning-disk confocal time-series of HeLa cells stably expressing DHC-GFP/CENP-A-mCherry and labeling of DNA with SiR-DNA undergoing the indicated treatments. The red arrow shows the presence of dynein in one KT in early metaphase. Relative time from NEB is represented. Time scale: hour:min. Scale bar: 10 μm. **b** DHC-GFP dynamics at KTs after NEB. Dots and dashed lines represent mean values and SD, respectively. The solid thick line represents the fitted curve. *N* (number of cells, number of independent experiments): siCTRL (13, 2), siCENP-E (14, 2). **c** Representative maximum intensity-projected point-scanning confocal images of U2OS WT/T422A monopoles immunostained against the indicated

proteins. Scale bar: 10 μm. **d** Fluorescence intensity profile plots of the indicated proteins (representing the region depicted by a white dashed line in **c**). **e** Quantification of spindle pole accumulation of indicated proteins in U2OS GFP-CENP-E WT and T422A cells represented in **c**. Values are plotted with mean ± SD. *N* (number of cells, number of independent experiments): GFP-CENP-E in WT (51, 3), GFP-CENP-E in T422A (55, 3), SPDLY in WT (54, 3), SPDLY in T422A (52, 3), ZW10 in WT (33, 3), ZW10 in T422A (34, 3), MAD1 in WT (43, 3), MAD1 in T422A (39, 3), BUBR1 in WT (50, 3), BUBR1 in T422A (49, 3). Statistical significance was determined by the Mann−Whitney *U*-test (unpaired, two-tailed; no normal distribution). *p* values are indicated.

independently of Spindly-bound dynein. In agreement with this hypothesis, polar chromosomes are observed in Spindly-depleted cells[25], and CENP-F and LIS1 were recently shown to be required to maintain a pool of KT-bound dynein after corona stripping and the establishment of end-on KT-MT attachments[12]. This dynein pool may be sufficient to move the chromosomes poleward in the absence of Spindly-CENP-E axis. Third, the poleward movement could be executed via MT dynamics. Corona removal may facilitate lateral-to-end-on attachment transition, leading to KT-MT depolymerization-based chromosome transport[46]. Once at the pole, these attachments could be released by the activity of Aurora A. Future work is required to explore these possibilities and to detail the interplay between the corona disassembly and chromosome poleward movement.

## Proposed mechanism for the role of AurA/B-CENP-E axis in kinetochore fibrous corona disassembly

Here, we propose a model in which AurA/B activity triggers the activation of CENP-E, enabling CENP-E to overcome the activity of dynein, and to consequently win the tug-of-war between these two motors of opposing directionality (Fig. 5f). During chromosome congression in early mitosis, when the peripheral polar chromosomes reach the pole[16], enhanced phosphorylation by AurA promotes CENP-E activity to direct the chromosomes toward the metaphase plate. This is further facilitated by MT-detyrosination, a tubulin post-translational modification associated with stable spindle MTs that promotes CENP-E processivity[47] and impedes the initiation of dynein movement[48]. Importantly, winning this tug-of-war allows CENP-E to keep the fibrous corona components at KTs until CENP-E becomes dephosphorylated by mitotic phosphatases, such as PP1[20,49]. Once CENP-E is inactivated by dephosphorylation, dynein transports its cargo toward the spindle poles, thereby disassembling the fibrous corona.

Our phospho-antibody-based data suggest that CENP-E dephosphorylation happens in a gradual manner, starting from the region most distal from the centromeric AurB. This would allow a stepwise removal of corona, with a smaller, centromere-proximal fraction being released the last, after chromosome biorientation and alignment in metaphase has been already achieved. Once at the pole, CENP-E is re-activated by AurA, which then allows the release of corona components from the pole. Thus, we have uncovered a mode to regulate CENP-E activity and propose a molecular mechanism underlying the fibrous corona disassembly in human cells.

During the revision of this article, three research groups reported independent preprint studies supporting the role of CENP-E in fibrous corona assembly[50–52]. Together with our findings that CENP-E activation by AurA/B controls fibrous corona disassembly, this opens an exciting direction for dissecting the mechanistic properties and function of the outermost KT layer.

## Methods
### Generation of CENP-E cell lines and cloning

To generate cells with inducible expression of CENP-E, we first integrated a Tet repressor (TetR) in U2OS cells by lentiviral transduction of pLenti CMV TetR Blast (716-1) vector (Addgene #17492). CENP-E WT

and T422A genes were commercially synthesized (GenScript) in a pBluescript II SK(-) vector. CENP-E was cloned using KpnI and BamHI into pENTR4-GFP (Addgene # 17396) vector for Gateway-compatible N-terminal GFP tagging. GFP-CENP-E constructs were later subcloned into pLenti CMV/TO puro DEST (Addgene #17293) and into a pΔT-DEST (gift from Stephan Geley, Innsbruck Medical University, Austria) destination vector by LR recombination (Invitrogen) for lentiviral transduction of U2OS TetR cells and for transient transfection, respectively. CENP-E W423A constructs were generated by site-directed mutagenesis of pENTR4-GFP vectors using the following primers: CENP-E W423A - F: 5′-attttgttaattttgccaaggcacgcagtaactcttcgttttcttttagcc-3′, R: 5′-ggctaaaagaaaacgaagagttactgcgtgccttggcaaaattaacaaaat-3; CENP-E T422A/W423A – F: 5′-attttgttaattttgccaaggcacgcagcaactcttcgttttcttttagc-3′, R: 5′-gctaaaagaaaacgaagagttgctgcgtgccttggcaaaattaacaaaat-3′.

Following infection, cells were selected with puromycin, and serial dilutions were performed for clonal selection. For experiments, endogenous CENP-E was depleted by transfection of 20 nM 3′UTR siRNA (5′ CCACUAGAGUUGAAAGAUA 3′)[20] 24 h before fixation/filming. Doxycycline (1 μg/ml, Sigma-Aldrich) was added overnight to induce GFP-CENP-E expression. Adenovirus encoding H2B-RFP (AV-H2B-RFP) were produced using pAd/CMV/V5-DEST Gateway Vector Kit (Thermo Fisher Scientific) according to the manufacturer's instructions. For live-cell imaging, cells were infected with AV-H2B-RFP for 24 h to enable DNA visualization and 20 nM SiR-tubulin + 5 μM verapamil (Spirochrome AG) was added for microtubule visualization.

For the generation of stable U2OS GFP-CENP-E/CENP-A-mCherry and HeLa DHC-GFP/CENP-A-mCherry cell lines, we performed lentiviral infection of the U2OS GFP-CENP-E (this study) and HeLa DHC-GFP (gift from Iain Cheeseman, Whitehead Institute for Biomedical Research, Cambridge, USA) cell lines using pLenti CENP-A-mCherry vector (Addgene #89767), followed by fluorescence activated cell-sorting (FACS) of mCherry-positive cells. DNA was labeled using 20 nM SiR-DNA (Spirochrome AG).

CENP-E fragments (1-859 and 1799-2701) encoding vectors were generated by PCR amplification with 5′ KpnI and 3′ BamHI flanking sites for initial cloning into a pENTR4-GFP vector. Next, CENP-E (1-859) and (1799-2701) sequences were subcloned by LR reaction into a pΔT-DEST-Myc and pΔT-FLAG-DEST vectors (gift from Stephan Geley, Innsbruck Medical University, Austria) respectively.

### Live cell imaging

Cells were seeded, transfected, and treated in glass bottom 35 mm dishes (MatTek). Time-lapse imaging was performed in a heated incubation chamber at 37 °C with controlled humidity and 5% CO2 supply using a Plan-Apochromat 63x/1.4NA oil objective with differential interference contrast mounted on an inverted Zeiss Axio Observer Z1 microscope (Marianas Imaging Workstation (3i−Intelligent Imaging Innovations, Inc.)), equipped with a CSU-X1 spinning disk confocal head (Yokogawa Corporation of America) and four laser lines (405, 488, 561, and 640 nm). Images were acquired using an iXon Ultra 888 EM-CCD camera (Andor Technology) and Slidebook 6.0.24 software (3i−Intelligent Imaging Innovations, Inc.) with 1μm separation interval between z-planes, covering the entire mitotic spindle. All

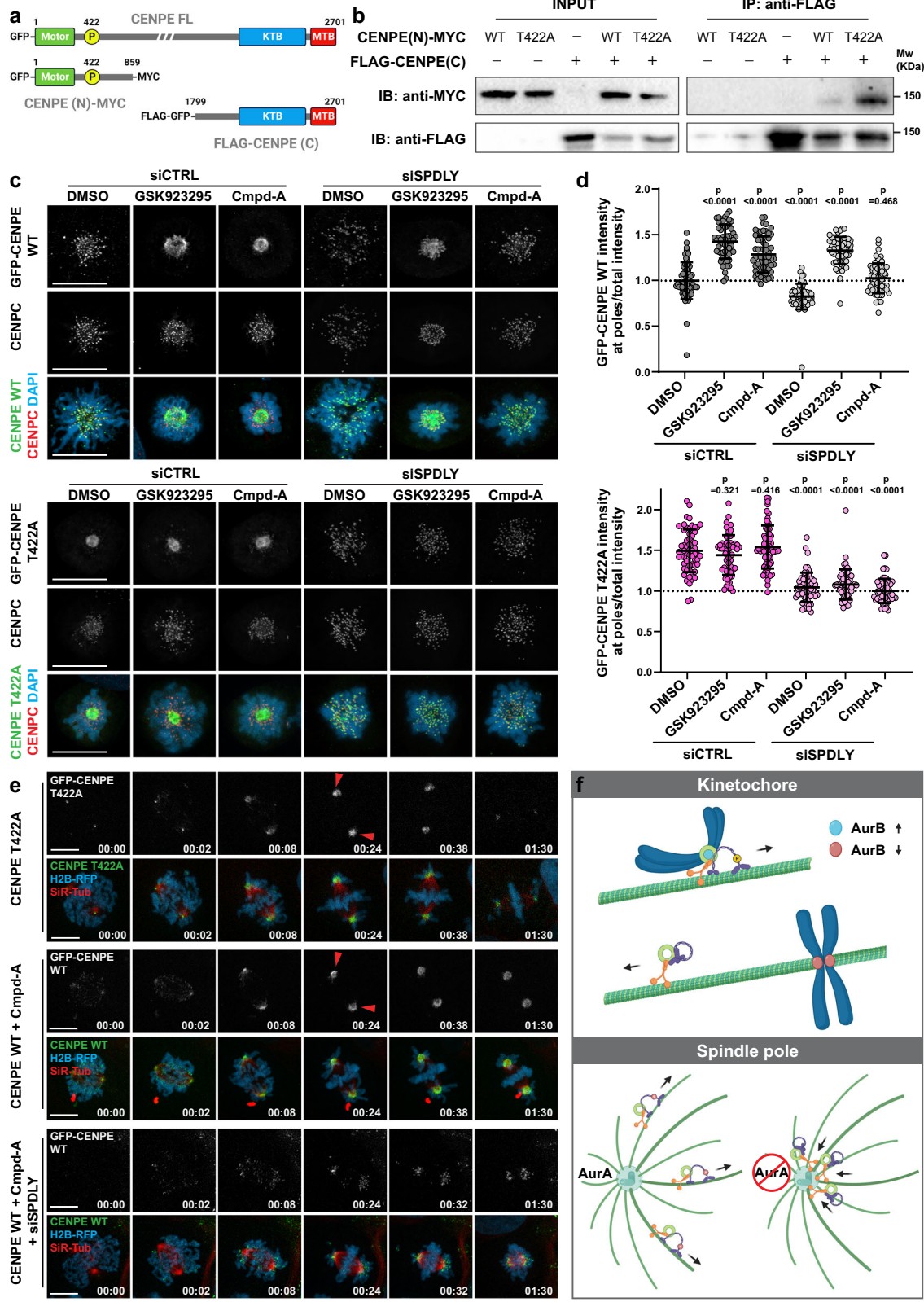

images represent the maximum-projections of z-stacks. Images were processed using ImageJ and Photoshop 2020 (Adobe).

To determine the SAC robustness, cells were treated with 100 nM nocodazole (Sigma) prior to imaging. Time-lapse imaging was performed in a heated incubation chamber at 37 °C with controlled humidity and 5% $CO_2$ supply, using an EC PlnN 10x /0,3 objective with differential interference contrast mounted on an inverted Zeiss Axio Observer Z1 microscope (Marianas Imaging Workstation from Intelligent Imaging and Innovations Inc. (3i), Denver, CO, USA), equipped with a X-Cite 110 LED white light source. Images were detected using ORCA-Flash4.0 v2 sCMOS camera (Hamamatsu). Six 3 μm-separated z-planes were collected every 15 min for 20 h. Time in mitosis was measured from mitotic rounding until post-mitotic flattening.

**Fig. 5 | Aurora A and B kinases activate CENP-E by releasing it from an auto-inhibited state. a** Schematic representation of the CENP-E constructs used in this study. **b** Co-immunoprecipitation of FLAG-GFP-CENP-E (1779-2701) construct with WT and T422A CENP-E-MYC (1-859). Four independent experiments showed similar results. **c** Representative maximum intensity-projected point-scanning confocal images of monopoles in U2OS CENP-E WT and T422A cell lines following the indicated treatments. Scale bar: 10 µm. **d** Quantification of GFP spindle pole/ total intensity ratio following the indicated treatments in CENP-E WT/T422A monopoles. Values are plotted with mean ± SD. *N* (number of cells, number of independent experiments) - GFP-CENP-E WT: siCTRL + DMSO (64, 3), siCTRL + GSK923295 (58, 3), siCTRL + Cmpd-A (59, 3), siSPDLY + DMSO (60, 3), siSPDLY + GSK923295 (59, 3), siSPDLY + Cmpd-A (61, 3); GFP-CENP-E T422A: siCTRL + DMSO (60, 3), siCTRL + GSK923295 (60, 3), siCTRL + Cmpd-A (61, 3), siSPDLY + DMSO (60, 3), siSPDLY + GSK923295 (60, 3), siSPDLY + Cmpd-A (58, 3). Note that in the absence of KT-bound dynein (CENP-E-WT + siSpindly) chromosomes are dispersed away from the pole by

polar ejection forces (PEFs)[16], lowering the ratio of polar and total intensity. Statistical significance was determined by the Mann–Whitney *U*-test (unpaired, two-tailed; no normal distribution). p values are indicated. **e** Spinning-disk confocal time-series of mitosis of GFP-CENP-E WT/T422A cells undergoing indicated treatments. Red arrowheads show spindle pole accumulation of non-phosphorylated CENP-E and inhibited CENP-E. Time: hour:min. Scale bar: 10 µm. **f** Illustrated model of the effects of CENP-E phosphorylation by Aurora A and B kinases. CENP-E activation is regulated by phosphorylation/dephosphorylation. Phosphorylation-activated CENP-E is dominant over dynein and thus facilitates chromosome congression to the spindle equator. CENP-E dephosphorylation inactivates CENP-E, thereby triggering dynein-mediated stripping of corona proteins toward the spindle pole. Re-activation of CENP-E by Aurora A phosphorylation at centrosome releases corona proteins from the spindle pole. Illustrations created with BioRender.com.

## Immunoflorescence
Cells were grown on 12 mm round cover glasses (Menzel Glaser) and fixed either by ice-cold methanol at -20 or pre-warmed 4% PFA in PHEM buffer (120 mM PIPES pH6,9, 50 mM HEPES, 20 mM EGTA, 8 mM MgSO4). For pT422 staining, cells were pre-extracted with 0.5% Triton X-100 (Sigma-Aldrich) in PHEM buffer for 30 s to be later fixed with 4% PFA in PHEM buffer with 100 nM Okadaic Acid (Tocris - Bio-Techne) for 10 min. The following primary antibodies were used in this study: mouse anti-CENP-E 1:500 (ab5093, Abcam), rabbit anti-CENP-E 1:500 (ab133583, Abcam), guinea-pig anti-CENP-C 1:2000 (PD030, MBL), rabbit anti-Astrin/MAP126 1:500 (A301-511A, Bethyl), rabbit anti-α-tubulin 1:500 (ab15246, Abcam), mouse-anti α-tubulin 1:2000 (T5168, clone B-5-1-2, Sigma), mouse anti-MAD1 1:200 (MABE867, Clone BB3-8, Millipore), goat anti-GFP 1:500 (600-101-215, Rockland), mouse anti-GFP 1:500 (sc-9996, Santa Cruz), mouse anti-Spindly 1:100 (H00054908, Abnova), mouse anti-FLAG-M2 1:1000 (f3165, Sigma), mouse anti-p150 1:1000 (610474, BD trans. lab), mouse anti-BUBR1 1:400 (gift from Jakob Nilsson, University of Copenhagen, Denmark), rabbit anti-ZW10 1:200 (ab21582, Abcam), rabbit anti-CENP-E pT422 1:200 (gift from Don Cleveland, University of California San Diego, USA). Goat anti-Mouse IgG (H + L) Highly Cross-Adsorbed Secondary Antibody, Alexa Fluor 488, 568, 645; Goat anti-Rabbit IgG (H + L) Highly Cross-Adsorbed Secondary Antibody, Alexa Fluor 488, 568, 645; Goat anti-Guinea Pig IgG (H + L) Highly Cross-Adsorbed Secondary Antibody, Alexa Fluor 568, 645; and Donkey anti-Mouse IgG (H + L) Highly Cross-Adsorbed Secondary Antibody, Alexa Fluor 488, 568, 647 (Invitrogen) secondary antibodies (Invitrogen) were used at 1:1000. For DNA counterstaining DAPI was used at 1 µg/ml (Sigma).

## Cell culture, transfection, and drug treatments
Human osteosarcoma U2OS cells (gift from S. Geley, Innsbruck Medical University, Innsbruck, Austria), U2OS-PA-GFP/mCherry-α-tubulin (gift from R. Medema, Netherlands Cancer Institute, Amsterdam, Netherlands), U2OS-EB1-GFP (gift from P. Draber, IMG ASCR, Prague, Czech Republic), U2OS Tet-On cell lines generated in this study (U2OS-GFP-CENPE: WT, T422A, W423A, T422A/W423A), HeLa Kyoto (Danish Cancer Society Research Center's Cell Line Bank), HeLa DHC-GFP (gift from I. Cheeseman, Whitehead Institute for Biomedical Research, Cambridge, USA), HeLa DHC-GFP/CENP-A mCherry (generated in this study), and HEK293T (gift from S. Geley, Innsbruck Medical University, Innsbruck, Austria) were grown in Dulbecco's Modified Eagle Medium (DMEM, Thermo Fisher) supplemented with 10% fetal bovine serum (FBS; Thermo Fisher). The immortalized human retinal epithelial cell line hTERT RPE-1 (ATCC) were grown in DMEM/F12 supplemented with 10% FBS. All cell lines were cultured at 37 °C in humidified conditions with 5% CO2.

For siRNA-mediated depletion experiments, cells were transfected in OptiMEM (Thermo Fisher) with Lipofectamine RNAiMAX (Thermo Fisher) with 40 nM siRNAs for 48 h – siSpindly: 5′-GAAAGGGU

CUCAAACUGAA-3′[25], siControl (non-targeting control siRNA): 5′- UGG UUUACAUGUCGACUAA-3′. For the rescue experiments, FLAG-tagged Spindly RNAi-resistant vectors were transfected using Xtreme-Gene for 54 h. 6 h after transfection fresh medium was added and cells were transfected with siRNAs.

For Aurora A and B kinase inhibition 250 nM MLN-8054 (Selleck Chemicals) and 4 µM ZM447439 (AstraZeneca) were added respectively to the cells 2 h before filming or fixation. For double Aurora A + B inhibition, cells were incubated with 250 nM MLN8054 for 2 h and 4 µM ZM447439 was added 30 min prior fixation. To depolymerize microtubules 3.3 µM nocodazole (Sigma) was added to the cell culture media 2 h before fixation/filming, except for CENP-E reloading experiments, where nocodazole was added immediately before live-cell imaging or 10 min before fixation, and for corona extension analysis, where cells were treated with nocodazole for 4 h. In all, 5 µM S-trityl-L-cysteine (STLC; Santa Cruz Biotechnology) was added to the cell culture media to induce monopolar spindles by inhibiting Eg5.

For CENP-E inhibition 200 nM GSK923295 (MedChemExpress) and 200 nM Cmpd-A (Takeda Pharmaceutical Company Limited) were added to the cells 10 min before imaging. In the case of fixed monopoles, inhibitors were added together with 5 µM of STLC 2 h before fixation. To measure the effect of CENP-E inhibitors in cells arrested in metaphase, GFP-CENP-E WT cells were treated with 5 µM of MG132 for 3 h, with or without adding 20 nM nocodazole 1 h prior to imaging.

## Image acquisition and quantification
For dynamic quantification of KT signal, live-cell imaging of U2OS GFP-CENP-E/CENP-A-mCherry and HeLa DHC-GFP/CENP-A-mCherry was performed. Frames corresponding to nuclear envelope breakdown (NEB) were selected and subsequent 18 frames imaged at 2 min intervals were used for quantification. Background subtracted KT intensities were measured using a 9×9 pixel square ROI for each KT using a custom-written script in Matlab (gift from A. Pereira, i3S Porto, Portugal). For each time point at least 20 KTs within each cell were analyzed. The average of mean KT intensities within a cell were plotted. Peak intensity after NEB was used to normalize subsequent time frames.

For quantification of GFP-CENP-E T422A reloading to KTs after MT depolymerization, cells in pseudometaphase were selected and 3.3 µM nocodazole was added immediately before filming. KT signal quantification was performed in U2OS GFP-CENP-E T422A CENP-A-mCherry cells as described above.

For quantification of KT intensity in fixed cells, images were acquired using a Zeiss AxioObserver Z1 wide-field microscope (×63 Plan-Apochromatic oil differential interference contrast objective lens, 1.4 NA) equipped with Metal halide arc lamp and Axiocam 702 mono CMOS camera and Zen 3.0 blue edition software (Carl Zeiss, Inc.). Representative images were acquired using LSM700 confocal microscope (Carl Zeiss Inc.) mounted on a Zeiss-Axio imager Z1 equipped with alpha-Plan-Apochromat 100x/1.46 oil DIC M27 and plan-apochromat 63×/1.40

oil DIC M27 objective (Carl Zeiss Inc.) and Zen 2010 B SP1 software (Carl Zeiss, Inc.). Intensities from at least 20 KTs per cell were measured. The fluorescence signal of the examined protein at each KT was normalized to CENP-C fluorescence. The protein of interest/CENP-C intensity ratios were subsequently normalized to control mean value.

The stripping velocity of GFP-CENP-E T422A was quantified manually using ImageJ from maximum intensity-projected images of early mitotic cells, as the distance traversed by the GFP comets over time.

To quantify centrosome/spindle pole intensities, a circular ROI covering the centrosomal area was drawn and cytoplasmic background subtraction was performed using the same ROI in Image-J. Mean intensities of each experiment was normalized to its control median, except for Fig. 5d where normalization was performed to control mean.

For corona volume quantification in HeLa cells and pT422 image acquisition, images were acquired using a Plan-Apochromat 63x/1.4 Oil DIC M27 objective at 30 °C mounted on an inverted ZEISS Axio Observer 7 microscope, coupled to an Elyra 7 super resolution system, and equipped with a Duolink sCMOS camera adapter for simultaneous two-color acquisition (Carl Zeiss, Inc., Oberkochen, Germany). 488 nm and 561 nm laser lines were used to illuminate samples and 9 (for corona volume quantification) or 13 (for pT422 image acquisition) phase-shifted images covering the mitotic plane were acquired. Images were later reconstructed with lattice SIM[2] module in Zen Black 3.0 SR FP2 Edition software. Corona volume was quantified using Image-J macro 3D object counter and unspecific signal coming from non-KT dots were eliminated by filtering out objects smaller than 0.02 µm[3].

pT422 and total CENP-E profile plots were measured by drawing a 1.5 µm line traversing the two kinetochores included in fully expanded coronas and using the RGB plot macro from Fiji/Image J.

3D surface rendering of total CENP-E and pT422 was done using Zen Blue 3.2 software.

## Microtubule flux and fluorescence intensity profiles analysis in metaphase cells

MT-flux rates in metaphase-arrested mitotic spindles were imaged and quantified using U2OS-PA-GFP/mCherry-α-tubulin cells[53] (gift from R. Medema, Netherlands Cancer Institute - NKI, Amsterdam, Netherlands). A transversal line-shaped region of interest, placed perpendicular to the main spindle axis on one side of the metaphase plate, was photoactivated by one 5 ms pulse from a 405 nm laser. Images were acquired every 5 s for 4 min over three 0.5 µm-separated z-planes. Kymograph generation and velocity quantification were performed using a custom-written Matlab script[54,55]. MT poleward flux rates were quantified by determining the slope of the fluorescence signal over time and using the spindle pole position as reference.

To quantify the distribution of GFP-CENP-E WT following inhibitor treatments along the metaphase-arrested mitotic spindle, the line scan function from ImageJ was used. Briefly, a 55-pixel wide line was drawn from the spindle pole to the metaphase plate. The mean intensities and half spindle lengths obtained from the scan were normalized by Min-max feature scaling. Quantifications were performed from images obtained 18 min after the addition of inhibitors. Statistical significance amongst the treatments was calculated for the normalized intensity values closest to the metaphase plate ($x = 1$).

## Immunoblotting and immunoprecipitation

For immunoblotting, cells following subjected treatments were collected and lysed in NP40 buffer (50 mM Tris−HCl pH 8, 150 mMNaCl, 5 mM EDTA, 0.5% NP-40, 1×EDTA-free protease inhibitor (Sigma-Aldrich), 1×phosphatase inhibitor cocktail (Sigma-Aldrich),1 mM PMSF). Protein extracts collected after centrifugation were subjected to SDS−PAGE and transferred onto nitrocellulose membrane (Bio-

Rad). The membranes were incubated with the following primary antibodies: mouse anti-vinculin 1:5000 (SAB4200729, Sigma-Aldrich), mouse anti-CENP-E 1:500 (sc-376685, Santa Cruz), goat anti-GFP 1:1000 (600-101-215, Rockland), mouse anti-p150 1:1000 (610474, BD Trans. Lab), rabbit anti-LIC 1:1000 (GTX120114, GeneTex), mouse anti-Spindly 1:1000 (H00054908-M01, Abnova), mouse anti-MYC 1:1000 (2276, clone 9B11, Cell Signaling), mouse anti-FLAG-M2 1:2000 (f3165, Sigma). HRP-conjugated secondary antibodies Peroxidase AffiniPure Goat Anti-Mouse, goat Anti-Rabbit, donkey anti-goat IgG (H + L) (Jackson ImmunoResearch) were used at 1:10,000 and visualized by ECL (Bio-Rad).

To immunoprecipitate full-length GFP-CENP-E, HEK 293 T cells were transfected with pCDNA3-GFP or pΔT-Dest-GFP-CENP-E for 48 h using metafectene (Biontex) following the manufacturer's instructions. The next day, cells were transfected either with control siRNA or CENP-E 3'UTR siRNA. 3.3 µM nocodazole was added 16 h prior collection to enrich the mitotic population. Cells were lysed in a detergent-free buffer (20 mM HEPES, 10 mM KCL, 1 mM MgCl2, 1 mM EDTA, 1 mM EGTA, 1×EDTA-free protease inhibitor (Sigma-Aldrich), 1×phosphatase inhibitor cocktail (Sigma-Aldrich), 1 mM PMSF) by 3 freeze/thaw cycles and sonication. To immunoprecipitate CENP-E C-terminal and N-terminal domains, HEK 293 T cells were transfected with pΔT-Dest-GFP-CENP-E-N-MYC constructs and/or pΔT-Dest-FLAG-GFP-CENP-E-C for 48 h. 3.3 µM nocodazole was added 16 h prior collection to enrich the mitotic population and enhance CENP-E-N-WT phosphorylation. Cells were lysed using a Triton-X-100-based buffer (25 mM HEPES, 100 mM NaCl, 1 mM EDTA, 10% (v/v) glycerol, 1% (v/v) Triton X-100, 1×EDTA-free protease inhibitor (Sigma-Aldrich), 1×phosphatase inhibitor cocktail (Sigma-Aldrich),1 mM PMSF) followed by sonication. The required antibodies for immunoprecipitation were bound to Dynabeads Protein G (Thermo Fisher) using a ratio of 1 µl antibody/10 µl of beads prior to the addition of cell lysates. After incubation with the lysate, Dynabeads were washed 3x with lysis buffer and proteins were eluted with SDS-Laemmli buffer and heating at 95 °C. All immunoblots and immunoprecipitation were reproduced at least 3 times. Unprocessed and uncropped blots are supplied in the Source Data file.

## Statistical analysis

All graphs and statistical analysis were generated in GraphPad Prism 8.0. Data points were tested for normality using D'Agostino & Pearson test. Accordingly, statistical significance was determined by Student's $t$ test (unpaired, two-tailed; normal distribution) or Mann−Whitney $U$-test (unpaired, two-tailed; no normal distribution). Details of the n values and statistical significance for each condition can be found in the figures and figure legends. For half-life calculations, raw data was fitted to single-phase exponential decay curves.

## Reporting summary

Further information on research design is available in the Nature Portfolio Reporting Summary linked to this article.

# Data availability

Source data are provided with this paper. Quantitative analysis datasets generated in this study are available as a Source Data file. Raw image data generated during this study are available freely for non-commercial research purposes from the corresponding author on request. Source data are provided with this paper.

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

## Acknowledgements

We thank Iain Cheeseman, Don Cleveland, Stephan Geley, Rene Medema, Ryo Mizojiri, Jakob Nilsson, Akihiro Ohashi, Antonio Pereira, and Benjamin Vitre for providing reagents and tools. We thank Carlos Conde, Stephan Geley, and Helder Maiato for critically reading the manuscript. We thank the groups of Geert Kops, Andrea Musacchio, and Julie Welburn for discussing the data. We thank Martina Barisic for exceptional technical assistance. This work was funded by the Danish Cancer Society (KBVU; R146-A9322, granted to M.B.), the Lundbeck Foundation (R215-2015-4081, granted to M.B.), and the Novo Nordisk Foundation (NNF19OC0058504, granted to M.B.).

## Author contributions

Design, data interpretation and manuscript preparation were carried out by S.E. and M.B.; S.E. performed and analyzed most of the experiments. M.K. and C.G.B. contributed to designing, analyzing, and validating experiments. G.R. performed and analyzed photoactivation experiments. Y.S. generated H2B-RFP adenovirus used in live cell imaging experiments. M.B. conceptualized and supervised the project.

## Competing interests

The authors declare no competing interests.
