## [Peer Review File · Nature Communications]

CENP-E activation by Aurora A and B controls kinetochore fibrous corona disassemblyREVIEWER COMMENTS

Reviewer #1 (Remarks to the Author):

In this manuscript, Eibes et al. addressed the role of CENP-E phosphorylation by Aurora A and B as a molecular switch to disassemble the fibrous corona, a crescent-shaped structure that transiently localizes to the outer kinetochore prior to the formation of the end-on attachment of kinetochores to microtubules. By revisiting the CENP-E phosphorylation site mutant at T422, which was previously shown to cause a chromosome congression defect, the authors found that this CENP-E mutant (CENP-E-T422A) is removed from --CENP-E at spindle poles and hold CENP-E at kinetochores, respectively. Removal of CENP-E-T422A from kinetochores was dependent on microtubules, while its recruitment to kinetochores was not, and this removal was mediated by dynein, as confirmed by the prevention of accumulation of the CENP-E mutant at spindle poles in cells depleted of Spindly, which recruits dynein to kinetochores. The authors found that CENP-E and dynein form a complex, and all other fibrous corona proteins accumulated at spindle poles together with CENP-E-T422A, demonstrating that the CENP-E phosphorylation at T422 is required to prevent polar accumulation of corona proteins. As a molecular basis for the role of the CENP-E phosphorylation at T422, the authors showed that the N-terminal and C-terminal regions of CENP-E bind together to form an autoinhibitory structure that is relieved by the T422 phosphorylation. The authors extended this finding by showing that chemical inhibition of CENP-E also resulted in its accumulation at spindle poles. Interestingly, treatment with GSK9223295, which induces rigor-binding of CENP-E to microtubules, promoted the accumulation of CENP-E even in the absence of dynein via microtubule poleward flux.

The reviewer enjoyed this well-focused and well-organized story. It is impressive that the authors have addressed the underlying mechanism of the previously-overlooked spindle pole localization of CENP-E when it is mutated at T422 or chemically inhibited. It was widely known that CENP-E inhibition causes chromosome congression defects, but it has not been recognized that it even delocalizes CENP-E from kinetochores. Overall, the authors' results and conclusions are convincing, except for the concerns listed below.

Major points

1. One of the remaining questions is whether the interaction between fibrous corona and kinetochore can be explained solely by tug-of-war between CENP-E and dynein. Fibrous corona carries kinetochores to and away from spindle poles by dynein and CENP-E, respectively, during prometaphase, whereas it is stripped away from kinetochores in metaphase. This transition in the mode of action is related to the conversion from lateral to end-on attachment of kinetochores to microtubules, which coincides with CENP-E dephosphorylation. However, the results in this manuscript suggest that even laterally-attached kinetochores are stripped of fibrous corona when CENP-E motor activity is inhibited. How then do kinetochores distinguish between poleward motion ('gliding') in early mitosis and fibrous corona stripping, both of which are mediated by dynein, which has won the tug-of-war against CENP-E? The authors should clarify whether chromosome poleward motion in early mitosis is suppressed and fibrous

corona stripping occurs instead when CENP-E motor activity is inhibited, and discuss the possibility that CENP-E motor activity is required to maintain fibrous corona on kinetochores even when it loses the tug-of-war.

2. Another related question is the integrity of fibrous corona. If the distribution of CENP-E-pT422 in Fig. 2h represents that a fraction of CENP-E closer to the centromeric region is phosphorylated, why is unphosphorylated CENP-E at the outer part not stripped away as shown in Fig. 2a? The authors should discuss the underlying mechanism.

3. It was previously shown that the CENP-E-T422 phosphorylation suppresses PP1 binding, which facilitates chromosome congression (Cell 142, 444-455, 2010). The authors should clarify how these two activities of the CENP-E-T422 phosphorylation, release from an autoinhibited state and disruption of PP1 binding, are related to corona stripping and chromosome congression. Additional CENP-E constructs containing mutations at the PP1-docking motif would discriminate the contribution of these two activities.

4. The claim that dynein and CENP-E form a complex (Fig. 3f) needs further clarification. It is already known that both of them are components of fibrous corona, but does the authors mean more direct interaction between them? If so, they need to show whether the interaction is direct or not, and if it is direct, which region of CENP-E and which components of the dynein complex interact.

5. Fibrous corona stripping is a mechanism to satisfy the spindle assembly checkpoint (SAC), and the authors' data suggest that fibrous corona proteins accumulate at spindle poles when CENP-E is depleted or inhibited. On the other hand, it is known that the SAC is activated and cells are arrested in mitosis in CENP-E-depleted or inhibited cells. The authors should show the kinetochore localization of the SAC proteins in CENP-E-depleted or inhibited cells and address whether the SAC is weakened in these cells.

Minor points

1. In Fig. 1d, it is interesting that the cells treated with MLN8054 showed a transient polar accumulation of CENP-E, and the reviewer agrees that the remaining KT-based activity of Aurora B is responsible for the phenotype. But why do the accumulation and release take time (~ 30 min) and occur sequentially even though Aurora B is active throughout the time course? Showing movies of GFP-CENP-E treated with STLC in the presence of an Aurora B-specific inhibitor or both Aurora A- and Aurora B-specific inhibitors would shed some light on this point.

2. In Fig. 1f, g, the authors should use an inhibitor more specific to Aurora B than ZM447439, which inhibits both Aurora A and Aurora B.

3. In Extended Data Fig. 3d, chromosome alignment was severely perturbed in cells expressing the FLAG-Spindly-F258A mutant compared to Spindly-depleted cells. Is this due to additional defects caused by the Spindly mutant in the presence of CENP-E-T422A? The authors need to clarify that point.

4. The localization of CENP-E-T422A and that of CENP-E in the presence of its inhibitors does not seem to be restricted to spindle poles, but shows some distribution around spindle poles. The authors should address whether CENP-E-T422A and chemically-inhibited CENP-E are indeed localized to spindle poles by comparing their distribution with that of the components of the pericentriolar material. If their distribution is not restricted to spindle poles, this should be clarified and discussed in the text: do they represent CENP-E-containing fragments that continuously move towards the spindle poles, or stopping somewhere in the middle on their way to spindle poles?

5. In Figure 2h, the result was interpreted as a fraction of CENP-E closer to inner centromere being preferentially phosphorylated. However, it could also represent the spatial distribution of phosphorylated T422 along the CENP-E molecule extending from inside to outside of kinetochores. This should be clarified by comparing the distribution of the phosphorylated CENP-E-T422 with that of the N-terminal and the C-terminal of CENP-E by labeling them with different markers or antibodies. This experiment would also be helpful to distinguish the autoinhibited state of CENP-E by observing the overlap of its N-terminal and C-terminal regions.

Reviewer #2 (Remarks to the Author):

CENP-E is a microtubule motor with essential functions in chromosome bi-orientation. Its regulation remains poorly understood. In this manuscript, Eibes and colleagues report an analysis of the role of a residue of CENP-E, Thr422, which has been previously implicated in the regulation of the interaction of CENP-E and protein phosphatase 1 (PP1) by Aurora kinases. The authors demonstrate that cells expressing CENP-E-T422A reached a pseudometaphase condition with uncongressed chromosomes and a very significant accumulation of CENP-E-T422A at spindle poles. This pole pool was prematurely removed from kinetochores through a Dynein-dependent mechanism, as the localization of CENP-E-T422A in nocodazole-treated cells, i.e. cells where microtubules had been depolymerised, was close to that of controls. With an assay exploiting monopolar spindles and specific small-molecule inhibitors, the authors provide convincing evidence that Aurora B is required to retain CENP-E-T422A at kinetochores, and that Aurora A is required for its mobilisation from spindle poles. Furthermore, imaging at high temporal resolution allowed the authors to visualise the movement of CENP-E-T422A towards the pole, and to measure its speed, shown to be consistent with Dynein transport. This was further confirmed by various additional analyses, including the depletion of the Dynactin-Dynein adaptor Spindly and a precipitation experiment showing an interaction between CENP-E and Dynactin-Dynein. Furthermore,

the authors report that CENP-E, while being dispensable for dynein loading to kinetochores, was required for its maintenance there, as cells depleted of CENP-E 'shed' dynein more rapidly than cells expressing CENP-E.

Collectively, this study advances our understanding of the regulation of CENP-E and of its role in the kinetochore corona. The experiments are properly controlled. From a mechanistic perspective, an important new piece of evidence is the intra-molecular interaction of the N- and C-terminal moieties and the suggestion that phosphorylation of Thr422 regulates it, as well as the physical interaction of CENP-E and Dynein. Sometimes the authors over-interpret their results or should refer more thoroughly to previous evidence, something that can be addressed textually.

Major points

-The last sentence of the abstract states that the current study establishes the mechanism of fibrous corona disassembly ('...long-sought mechanism of fibrous corona disassembly'. This is then repeated at the end of page 8, where the authors write 'Thus, these experiments demonstrate that CENP-E phosphorylation by AurA/B delays corona stripping. Finally, it is repeated as a final statement on page 12: 'Thus, we have uncovered a novel mode of regulation of CENP-E activity and identified the molecular mechanism underlying the fibrous corona disassembly in human cells.'

This conclusion has to be expressed with more caution, as the authors only provide indirect evidence in its support. (They also gloss over previous evidence on the mechanistic basis of the regulation of T422, something that I address in my next comment.) Specifically, this study establishes a correlation between the phosphorylation state of T422 and chromosome congression, partly recapitulating work in reference 19 (Kim, Holland et al. Cell 2010; work by the laboratory of Don C. Cleveland). However, there is no evidence in this paper that persisting phosphorylation of T422 prevents corona disassembly in presence of bi-oriented chromosomes, which is what would be required to support the two statements above. In my perception, the data presented here imply that lack of 422 phosphorylation (obtained with the T422A mutant) results in premature corona disassembly, while corona assembly is not overtly affected. This however only shows that the corona is less stable if this residue cannot be phosphorylated, not that its dephosphorylation is the crucial trigger of corona disassembly. Furthermore, the authors clearly indicate that the T422A mutation also prevents retention of CENP-E after corona disassembly, thus suggesting that phosphorylation of this residue may affect CENP-E beyond its mere participation in the corona. In conclusion, the authors' observations are compatible with the claim, but the authors do not prove it, nor do they detail the mechanism of corona disassembly, something on which I comment below. The three sentences should be modified accordingly.

-On page 9, the authors claim that '...the impact of phosphorylation on intra- or inter-molecular properties of CENP-E remains unknown'. This should be revised. Phosphorylation of CENP-E on residue

422 by Aurora kinases has been described previously (reference 19), and convincingly linked to the recruitment of PP1 phosphatase. This previous evidence contradicts the statement, because the previous work establishes an 'inter-molecular' property of CENP-E. In the current manuscript, lack of phosphorylation is shown to lead to a stable intramolecular interaction of the N- and C-terminal moieties of CENP-E, in turn proposed to inhibit the motor domain. The two models are not incompatible, because the intramolecular interaction demonstrated here may require dephosphorylation of additional phosphorylation sites, a hypothesis that the author do not discuss but that is in line with work by Espeut et al. (Mol Cell 2010), who demonstrated that CENP-E auto-inhibition can be relieved by MPS1 and CDK1 phosphorylation of the CENP-E C-terminal tail. A CENP-E intramolecular interaction was also discussed in Vitre et al. (MBoC 2014), where additional segments of the CENP-E kinetochore-targeting domain were shown to inhibit motor activity if positioned too closely to the motor domain.

Thus, I suggest that this sentence, and the following discussion, should be revised to give proper credit to the substantial existing evidence that CENP-E may exist in an auto-inhibited state. I also re-iterate that the present study advances, but does not unveil, the mechanism of corona disassembly, as this may be much more complex than implied by the study of a single phosphorylation site. Detailing the mechanism would require substantial additional work.

-Related to the comments above, also on page 9 the authors write 'In the absence of T422 phosphorylation, the N-terminal domain co-immunoprecipitated stronger with the C-terminal domain, indicating that non-phosphorylated CENP-E has a higher predisposition to be in an auto-inhibited state". In fact, what is being shown is an intra-molecular interaction, not an auto-inhibition, which is rather a hypothesis.

-On page 10, the authors write 'This suggests that the C-terminal fragment interacts and inhibits endogenous CENP-E, which is in line with a dominant negative effect caused by the expression of different C-terminal KT-targeting domain constructs of CENP-E'. Again, this seems to be the preferred argument, but there are plausible alternatives to it. The C-terminal region of CENP-E expressed by the authors to demonstrate the dominant-negative effect they describe is the kinetochore-targeting domain, which may also exercise its dominant-negative effects through displacement of endogenous CENP-E from kinetochores and from Dynein, and therefore not by inhibiting endogenous CENP-E. This possibility should be discussed in the text.

- 'Our phospho-antibody-based data suggest that CENP-E dephosphorylation happens in a gradual manner, starting from the region most distal from the centromeric AurB'. What is the evidence for this? The levels of P-T422 (Figure 2h) are expected to be maximal in nocodazole. What is the evidence that there is any dephosphorylated CENP-E at kinetochores under these conditions? Also, please note that the P-T422 signal seems to be internal even to CENP-C in these images, which is possible but counterintuitive. So, I would recommend caution in the interpretation of these images.

Minor points

Abstract

- 'long-sought mechanism' should possibly read 'long-sought-for mechanism'
- There are various instances of 'This data...' – As 'data' is plural, the authors should write 'These data...'
- There is a bit of a disconnect in the manuscript between the first experiments demonstrating a role of Dynein in CENP-E 'stripping', described on page 5, and other experiments on the same topic discussed on page 7. These experiments are also described in the context of different figures, and I wonder if the narrative wouldn't benefit from coupling the description of these experiments.
- '...toroidal allocation...' – Please consider '...radial distribution...' as an alternative
- '...either of inhibitors...' Consider '...either inhibitor...'
- Throughout the manuscript, please consider removing adverbs like 'intriguingly', 'strikingly' and related adjectives. Their use is highly subjective and unnecessary.

Reviewer #3 (Remarks to the Author):

Kinetochores are large protein machines that drive cell division by forming complex, force-generating attachments to spindle microtubules. These functionalities are mediated by several kinetochore sub-domains, with the distal fibrous corona domain being the least well characterised. The fibrous corona is the most dynamic region of the kinetochore, expanding into a supramolecular meshwork before being disassembled, in part, by the dynein transport of corona cargoes along attached microtubules. Characterising corona disassembly is crucial for our understanding of how kinetochores permit accurate and timely cell division, an event that is essential for the growth, development and homeostasis of metazoans.

In this study, Eibes and colleagues have used engineered cell lines, live-cell imaging, siRNA-mediated depletion and molecular inhibition to show how an Aurora A-Aurora B-CENP-E axis controls chromosome congression and corona disassembly by dynein. Given the importance of corona disassembly for mitotic progression, and the scarcity of information on how it is regulated, this study represents an important advance of broad interest. Overall, the experiments are well performed and controlled, the conclusions are supported by the data and the manuscript is clearly written. Therefore, I support publication of this study in Nature Communications after the following revisions.

1. Can the authors clarify what data were used to calculate p-values? In some instances visually small differences have incredibly small p-values, which suggests to me that the number of kinetochores, rather than the number of independent experiments, have been used for the calculation. It would be helpful if the average for each biological replicate was added to the presented plots.

2. I am concerned about the accuracy of the images and associated quantification presented in figure 2h. In order to conclude that CENP-E and CENP-E pT422 are spatially separated I suggest the authors include at least two of the following. (1) Higher resolution imaging like the SIM data presented in Extended data figure 2b, (2) quantify a much larger sample and/or (3), plot distance measurements with error bars and associated statistics.

3. Kinetochores are hard to distinguish in the images presented in Figure 4c. Can the authors include some zooms so that the staining patterns are clearer to the reader? Furthermore, can the authors comment on why the CENP-E WT signal shows a different spatial distribution between conditions. For example, CENP-E WT appears to localise further from the monopole in the spindly panel when compared to the p150 panel.

REVIEWER COMMENTS

Reviewer #1 (Remarks to the Author):

In this manuscript, Eibes et al. addressed the role of CENP-E phosphorylation by Aurora A and B as a molecular switch to disassemble the fibrous corona, a crescent-shaped structure that transiently localizes to the outer kinetochore prior to the formation of the end-on attachment of kinetochores to microtubules. By revisiting the CENP-E phosphorylation site mutant at T422, which was previously shown to cause a chromosome congression defect, the authors found that this CENP-E mutant (CENP-E-T422A) is removed from \rightarrow CENP-E at spindle poles and hold CENP-E at kinetochores, respectively. Removal of CENP-E-T422A from kinetochores was dependent on microtubules, while its recruitment to kinetochores was not, and this removal was mediated by dynein, as confirmed by the prevention of accumulation of the CENP-E mutant at spindle poles in cells depleted of Spindly, which recruits dynein to kinetochores. The authors found that CENP-E and dynein form a complex, and all other fibrous corona proteins accumulated at spindle poles together with CENP-E-T422A, demonstrating that the CENP-E phosphorylation at T422 is required to prevent polar accumulation of corona proteins. As a molecular basis for the role of the CENP-E phosphorylation at T422, the authors showed that the N-terminal and C-terminal regions of CENP-E bind together to form an autoinhibitory structure that is relieved by the T422 phosphorylation. The authors extended this finding by showing that chemical inhibition of CENP-E also resulted in its accumulation at spindle poles. Interestingly, treatment with GSK9223295, which induces rigor-binding of CENP-E to microtubules, promoted the accumulation of CENP-E even in the absence of dynein via microtubule poleward flux.

The reviewer enjoyed this well-focused and well-organized story. It is impressive that the authors have addressed the underlying mechanism of the previously-overlooked spindle pole localization of CENP-E when it is mutated at T422 or chemically inhibited. It was widely known that CENP-E inhibition causes chromosome congression defects, but it has not been recognized that it even delocalizes CENP-E from kinetochores. Overall, the authors' results and conclusions are convincing, except for the concerns listed below.

We are grateful to the reviewer for recognizing the quality and importance of our work.

Major points:

1. One of the remaining questions is whether the interaction between fibrous corona and kinetochore can be explained solely by tug-of-war between CENP-E and dynein. Fibrous corona carries kinetochores to and away from spindle poles by dynein and CENP-E, respectively, during prometaphase, whereas it is stripped away from kinetochores in metaphase. This transition in the mode of action is related to the conversion from lateral to end-on attachment of kinetochores to microtubules, which coincides with CENP-E dephosphorylation. However, the results in this manuscript suggest that even laterally-attached kinetochores are stripped of fibrous corona when CENP-E motor activity is inhibited. How then do kinetochores distinguish between poleward motion ('gliding') in early mitosis and fibrous corona stripping, both of which are mediated by dynein, which has won the tug-of-war against CENP-E? The authors should clarify whether chromosome poleward motion in early mitosis is suppressed and fibrous corona stripping occurs instead when CENP-E motor activity is inhibited, and discuss the possibility that CENP-E motor activity is required to maintain fibrous corona on kinetochores even when it loses the tug-of-war.

We thank the reviewer for raising this important question. Our results suggest that corona stripping is initiated by lateral attachment when CENP-E is inactivated/non-phosphorylated. We detect corona stripping in very early prometaphase in T422A cell lines and CENP-E inhibited cells. This is in line with a recent study showing that the dynein-mediated removal of corona proteins does not require stable end-on kinetochore-microtubule attachments (Ide et al Mol Biol Cell 2023, PMID: 37126397).

Regarding the “gliding vs stripping” paradox, we do observe a poleward movement of chromosomes in CENP-E T422A or CENP-E-inhibited cells, resulting in chromosomes being trapped in the vicinity of the spindle poles, as it has been shown upon CENP-E depletion. In the revised version of our manuscript, we discuss few possibilities that could explain this poleward movement in cells with non-phosphorylated/inactivated CENP-E:

1) Using quantitative high temporal resolution imaging of CENP-E T422A dynamics at kinetochore, we observe corona stripping in early mitosis, suggesting that lateral attachments are sufficient for the initiation of corona stripping. Since elimination of dynein and CENP-E should decrease the stability of lateral attachments, a frequent transition from lateral to unattached state is expected. As presented in Figure 2d and 2e, when microtubule attachment is disrupted, CENP-E T422A reloads (presumably together with the rest of corona proteins) to kinetochore, which could re-establish the lateral attachment. This loop of corona expansion, lateral attachment, corona stripping and detachment could be sufficient for gradual poleward movement (please see the illustration below).

2) Chromosome poleward movement may be triggered independently of Spindly-bound dynein. Even though elimination of the dynein adaptor Spindly prevents the recruitment of most of the kinetochore dynein, polar chromosomes are present in Spindly depleted cells, supporting the idea of an alternative, Spindly-independent, mechanism of dynein-mediated poleward movement. A recent study has shown that CENP-F and LIS1 are required to maintain a pool of dynein at the kinetochore after corona stripping and the establishment of end-on attachments (Auckland et al, J Cell Biol 2020 PMID: 32207772). This pool of kinetochore dynein may be sufficient to drive the poleward movement in the absence of Spindly-CENP-E axis.

3) Another complementary possibility is that the poleward movements are directed by microtubule dynamics. Chromosomes may be transported towards the poles by kinetochore-microtubule depolymerization resulting from premature end-on attachment upon corona removal. Once the chromosomes reach the pole, these attachments could be released by the activity of Aurora A.

Although this is certainly an important question, we believe that addressing it in full would require a substantial set of experiments, and is therefore out of the scope of this manuscript and its current revision.

2. Another related question is the integrity of fibrous corona. If the distribution of CENP-E-pT422 in Fig. 2h represents that a fraction of CENP-E closer to the centromeric region is phosphorylated, why is unphosphorylated CENP-E at the outer part not stripped away as shown in Fig. 2a? The authors should discuss the underlying mechanism.

To allow the full expansion of fibrous corona displayed in Figure 2h, we completely depolymerized microtubules using 3.3 μM nocodazole. Because of the absence of microtubules, the dynein-mediated stripping of corona proteins is not possible under these conditions.

3. It was previously shown that the CENP-E-T422 phosphorylation suppresses PP1 binding, which facilitates chromosome congression (Cell 142, 444-455, 2010). The authors should clarify how these two activities of the CENP-E-T422 phosphorylation, release from an autoinhibited state and disruption of PP1 binding, are related to corona stripping and chromosome congression. Additional CENP-E constructs containing mutations at the PP1-docking motif would discriminate the contribution of these two activities.

We thank the reviewer for this important suggestion. The work done by Kim, Holland et al (Cleveland lab; Cell 142, 444-455, 2010) showed that T422 phosphorylation and PP1 binding are two connected events. The PP1-docking motif overlaps with the Aurora A/B phosphorylation motif, and CENP-E phosphorylation weakened the binding of PP1. We agree with the reviewer that it is important to address a possible role of PP1 during CENP-E inhibition, as PP1 binding may be required for the structural changes leading to CENP-E undertaking an “open” or “closed” conformation. Another possibility is that PP1 binding is required for the following dephosphorylation of other putative phosphorylations happening in the CENP-E C-terminal tail (Espeut et al, Mol Cell 2008, PMID: 18342609) to promote motor inhibition. In the revised version of our manuscript, we generated two new stable cell lines containing a point mutation of the PP1 docking motif: CENP-E W423A and CENP-E T422A/W423A (new Supplementary Fig. 5). To avoid any interference with the Aurora kinases phosphorylation site, and because W423A was shown to be sufficient to block PP1 binding to Xenopus N-terminal CENP-E (Cell 142, 444-455, 2010), we used this single point mutant of the PP1 docking motif.

If CENP-E-T422A promoted the autoinhibition via PP1 binding, CENP-E-T422A/W423A is expected to rescue the phenotype observed upon the expression of CENP-E-T422A. Live-cell imaging of U2OS cells expressing a single point mutant GFP-CENP-E-W423A displayed localization and behavior similar to GFP-CENPE-WT (new Supplementary Fig. 5). On the other hand, a double point mutant GFP-CENP-E-T422A/W423A-expressing cells arrested in mitosis, displaying chromosome congression problems and polar accumulation of GFP-CENPE-T422A/W423A, thus failing to rescue the phenotype induced by GFP-CENPE-T422A expression (new Supplementary Fig. 5). This suggests that the absence of T422 phosphorylation does not inhibit CENP-E by promoting the binding of PP1.

4. The claim that dynein and CENP-E form a complex (Fig. 3f) needs further clarification. It is already known that both of them are components of fibrous corona, but does the authors mean more direct interaction between them? If so, they need to show whether the interaction

is direct or not, and if it is direct, which region of CENP-E and which components of the dynein complex interact.

In the Figure 3f, we show that CENP-E co-immunoprecipitates with different components of dynein/dynactin complex. From this experiment, we cannot conclude whether the interaction between CENP-E and dynein is direct or indirect, and that has not been our intention. However, we still find this experiment informative because, although CENP-E and dynein are known to colocalize at the fibrous corona, to our knowledge their interaction (either direct or indirect) has not been previously shown. During the revision of this paper, a preprint study reported a direct interaction between CENP-E and Rod-ZW10-Zwilch-Spindly complex (Cmentowski et al, bioRxiv 2023, PMID: 37163019), which could indirectly mediate the formation of a complex containing CENP-E and dynein. Still, independently of the data supporting the indirect interaction, a potential co-existence of direct interaction cannot be excluded.

5. Fibrous corona stripping is a mechanism to satisfy the spindly assembly checkpoint (SAC), and the authors' data suggest that fibrous corona proteins accumulate at spindle poles when CENP-E is depleted or inhibited. On the other hand, it is known that the SAC is activated and cells are arrested in mitosis in CENP-E-depleted or inhibited cells. The authors should show the kinetochore localization of the SAC proteins in CENP-E-depleted or inhibited cells and address whether the SAC is weakened in these cells.

The relationship between the corona removal and SAC silencing is an important question and we thank the reviewer for motivating us to clarify it.

Although leading to a premature stripping of corona components (including the SAC proteins), CENP-E T422A expression or chemical inhibition of CENP-E results in uncongressed polar chromosomes that are not end-on attached to microtubules, as evident from our Astrin stainings displayed in Supplementary Fig. 1a, b. This is similar to CENP-E immunodepleted cells (McEwen et al, Mol Cell Biol 2001, PMID: 11553716). The unattached kinetochores can reload the corona/SAC proteins, as we show in Figure 2d, e, thereby keeping the SAC active. A recent publication (Ide et al, Mol Biol Cell 2023 PMID: 37126397) proposes that dynein-mediated corona stripping primes the phosphatase-dependent SAC silencing that responds to stable end-on microtubule attachments. This places the phosphatase pathway as the final step of SAC silencing, which could keep the SAC active even when corona is removed due to CENP-E T422A expression or chemical inhibition of CENP-E.

Following the reviewer's comments, we addressed this issue in more detail. First, we immunostained MAD1 to address whether the CENP-E T422A-induced polar chromosomes can keep the SAC active. Although the kinetochores of polar chromosomes strongly colocalize with the spindle pole-accumulated MAD1, we could detect MAD1 at the kinetochores of polar chromosomes when they are slightly separated from the spindle pole (new Supplementary Fig. 1a).

Next, we challenged the SAC robustness by measuring the duration of mitosis upon addition of 100nM Nocodazole. We show that CENP-E WT and T422A cells exit mitosis within a similar time frame, indicating that both cell lines have comparable SAC strength (new Supplementary Fig. 1c).

Finally, to prove that the SAC in CENP-E-inactivated cells is active due to the presence of unattached polar chromosomes, we inhibited the anaphase-promoting complex (APC) using APCin and ProTAME, which arrested CENP-E WT cells in metaphase, after allowing all chromosomes to become bi-oriented and congressed. Then we washed out APCin/ProTAME to allow the cells to enter anaphase with Cmpd-A-inhibited CENP-E. Although Cmpd-A induced the accumulation of CENP-E WT at the spindle poles, these cells exited mitosis in a similar way as control cells, indicating that CENPE-inactivated cells remain arrested in mitosis due to the presence of polar chromosomes (please see the figure below).

Minor points:

1. In Fig. 1d, it is interesting that the cells treated with MLN8054 showed a transient polar accumulation of CENP-E, and the reviewer agrees that the remaining KT-based activity of Aurora B is responsible for the phenotype. But why do the accumulation and release take time (~ 30 min) and occur sequentially even though Aurora B is active throughout the time course? Showing movies of GFP-CENP-E treated with STLC in the presence of an Aurora B-specific inhibitor or both Aurora A- and Aurora B-specific inhibitors would shed some light on this point.

Our data suggest that the efficacy of Aurora B to release CENP-E from the spindle poles in Aurora A-inhibited cells depends on the proximity of the kinetochores to the spindle pole. To generate an effect on the pole-accumulated CENP-E, the kinetochore-localized Aurora B must come close to the spindle pole. This dependence between the chromosome/kinetochore positioning and CENP-E release is visible in Figure 1d. Below please see a version of the

Figure 1d with an extra panel showing DNA alone. Here it is visible that chromosomes surround the pole ~27 minutes after NEB.

Following the reviewer's advice, we have also added time-lapse images of CENP-E WT treated with STLC in the presence of Aurora B inhibitor (new Supplementary Fig. 1e). In line with our data in Figure 1f, Aurora B inhibition in monopoles did not interfere with CENP-E loading to kinetochores, but rather induced faster kinetochores stripping (new Supplementary Fig. 1e) compared to control STLC-treated cells (Figure 1d).

2. In Fig. 1f, g, the authors should use an inhibitor more specific to Aurora B than ZM447439, which inhibits both Aurora A and Aurora B.

Although the initial study from Stephen Taylor's lab reported that ZM447439 equally inhibited Aurora A and B, with IC50 values of ~100 nM (Ditchfield et al, J Cell Biol 2003, PMID: 12719470), a follow-up study from the same group revealed that ZM447439 inhibited Aurora A and B with IC50 values of 1000 nM and 50 nM respectively (Girdler et al, J Cell Sci 2006, PMID: 16912073). The in vitro assays performed in the latter study used ATP concentrations closer to physiological levels, which is likely the reason behind the observed discrepancy. The evidence provided in Girdler et al study that ZM447439 is a potent Aurora B inhibitor that is much less effective against Aurora A (20-fold difference), explains why neither the initial study nor any of the following studies, including our manuscript, observed any Aurora A inhibition-like phenotypes in cells treated with ZM447439. This is the reason why ZM447439 is still widely used as a selective Aurora B inhibitor, and we believe that it justifies its usage in our study.

3. In Extended Data Fig. 3d, chromosome alignment was severely perturbed in cells expressing the FLAG-Spindly-F258A mutant compared to Spindly-depleted cells. Is this due to additional defects caused by the Spindly mutant in the presence of CENP-E-T422A? The authors need to clarify that point.

By analyzing different Spindly-F258A mutant and Spindly-depleted cells, we have not observed a stronger effect of the Spindly mutant in the presence of CENP-E T422A. However, we agree with the reviewer that the selected picture could drive to that conclusion. To avoid that, we have added an additional example of the FLAG-Spindly-F258A mutant-expressing cell (new Supplementary Fig. 3d). Although the strength of the phenotype in these

two examples is slightly different, it is evident that this mutant prevents the stripping and polar accumulation of GFP-CENP-E T422A.

4. The localization of CENP-E-T422A and that of CENP-E in the presence of its inhibitors does not seem to be restricted to spindle poles, but shows some distribution around spindle poles. The authors should address whether CENP-E-T422A and chemically-inhibited CENP-E are indeed localized to spindle poles by comparing their distribution with that of the components of the pericentriolar material. If their distribution is not restricted to spindle poles, this should be clarified and discussed in the text: do they represent CENP-E-containing fragments that continuously move towards the spindle poles, or stopping somewhere in the middle on their way to spindle poles?

As shown in different figures throughout the manuscript, and as the reviewer points out, CENP-E and other corona components seem to be accumulated surrounding the spindle pole, as it has been seen in other publications using the ATP reduction assays (e.g. Gassmann et al, *Genes Dev*, 2010, PMID: 20439434; Famulski et al, *PLoS One* 2011, PMID: 21305043). As spindle pole accumulation, we consider the accumulation at the minus-ends of microtubules emerging from centrosomes, since no centrosomal accumulation is detected in the absence of polymerized microtubules. Below we show a SIM image of an early prometaphase of a T422A cell stained against GFP CENP-E T422A and pericentrin that further supports our observations.

5. In Figure 2h, the result was interpreted as a fraction of CENP-E closer to inner centromere being preferentially phosphorylated. However, it could also represent the spatial distribution of phosphorylated T422 along the CENP-E molecule extending from inside to outside of kinetochores. This should be clarified by comparing the distribution of the phosphorylated CENP-E-T422 with that of the N-terminal and the C-terminal of CENP-E by labeling them with different markers or antibodies. This experiment would also be helpful to distinguish the autoinhibited state of CENP-E by observing the overlap of its N-terminal and C-terminal regions.

To localize phosphorylated CENP-E in fully expanded corona, we induced complete depolymerization of microtubules via high dose of nocodazole. Under these conditions, it is not expected to detect intramolecular differences in CENP-E localization. These differences could be expected only upon the microtubule binding, which would likely cause the stretching of CENP-E, with the N-terminal/motor domain binding the microtubules (outer) and the C-terminal domain binding the kinetochore (inner).

Still, following the reviewer's suggestion, we analyzed this possibility. We see that p422 CENP-E is more centromeric compared to total CENP-E. Since the T422 phosphorylation is in the proximity of the motor domain, the antibodies against this phosphorylation bind to the N-terminal domain of CENP-E. The CENP-E antibody we used for immunofluorescence (referred in the old Figure 2h as "total CENP-E") was generated against the full-length recombinant protein, without specifying the epitope. To define the binding sites of the two commercial antibodies available in our lab, the mouse CENP-E (1H12; Abcam ab5093; our "total CENP-E") and the rabbit monoclonal CENP-E (Abcam ab133583), we performed immunoblotting using our truncated CENP-E constructs covering the N- and C-terminal fragments of CENP-E. We detected that the mouse CENP-E antibody preferably recognized the C-terminal region of CENP-E, whereas the rabbit CENP-E antibody bound to the N-terminal region (please see the figure below). We used these two antibodies to study the possibility of intramolecular differences in localization under the same conditions we used for p422 stainings. As expected, due to the absence of microtubules (as discussed above), we detected a strong co-localization of these two antibodies, suggesting that the differences we observe in the localization of phosphorylated and non-phosphorylated CENP-E are not due to different intramolecular localization.

Reviewer #2 (Remarks to the Author):

CENP-E is a microtubule motor with essential functions in chromosome bi-orientation. Its regulation remains poorly understood. In this manuscript, Eibes and colleagues report an analysis of the role of a residue of CENP-E, Thr422, which has been previously implicated in the regulation of the interaction of CENP-E and protein phosphatase 1 (PP1) by Aurora kinases. The authors demonstrate that cells expressing CENP-E-T422A reached a pseudometaphase condition with uncongressed chromosomes and a very significant accumulation of CENP-E-T422A at spindle poles. This pole pool was prematurely removed from kinetochores through a Dynein-dependent mechanism, as the localization of CENP-E-T422A in nocodazole-treated cells, i.e. cells where microtubules had been depolymerised, was close to that of controls. With an assay exploiting monopolar spindles and specific small-molecule inhibitors, the authors provide convincing evidence that Aurora B is required to retain CENP-E-T422A at kinetochores, and that Aurora A is required for its mobilisation from spindle poles. Furthermore, imaging at high temporal resolution allowed the authors to visualise the movement of CENP-E-T422A towards the pole, and to measure its speed, shown to be consistent with Dynein transport. This was further confirmed by various additional

analyses, including the depletion of the Dynactin-Dynein adaptor Spindly and a precipitation experiment showing an interaction between CENP-E and Dynactin-Dynein. Furthermore, the authors report that CENP-E, while being dispensable for dynein loading to kinetochores, was required for its maintenance there, as cells depleted of CENP-E 'shed' dynein more rapidly than cells expressing CENP-E.

Collectively, this study advances our understanding of the regulation of CENP-E and of its role in the kinetochore corona. The experiments are properly controlled. From a mechanistic perspective, an important new piece of evidence is the intra-molecular interaction of the N- and C-terminal moieties and the suggestion that phosphorylation of Thr422 regulates it, as well as the physical interaction of CENP-E and Dynein. Sometimes the authors over-interpret their results or should refer more thoroughly to previous evidence, something that can be addressed textually.

We thank the reviewer for appreciating the mechanistic advances provided by our study.

Major points:

-The last sentence of the abstract states that the current study establishes the mechanism of fibrous corona disassembly ('...long-sought mechanism of fibrous corona disassembly'. This is then repeated at the end of page 8, where the authors write 'Thus, these experiments demonstrate that CENP-E phosphorylation by AurA/B delays corona stripping. Finally, it is repeated as a final statement on page 12: 'Thus, we have uncovered a novel mode of regulation of CENP-E activity and identified the molecular mechanism underlying the fibrous corona disassembly in human cells.'

This conclusion has to be expressed with more caution, as the authors only provide indirect evidence in its support. (They also gloss over previous evidence on the mechanistic basis of the regulation of T422, something that I address in my next comment.) Specifically, this study establishes a correlation between the phosphorylation state of T422 and chromosome congression, partly recapitulating work in reference 19 (Kim, Holland et al. Cell 2010; work by the laboratory of Don C. Cleveland). However, there is no evidence in this paper that persisting phosphorylation of T422 prevents corona disassembly in presence of bi-oriented chromosomes, which is what would be required to support the two statements above. In my perception, the data presented here imply that lack of 422 phosphorylation (obtained with the T422A mutant) results in premature corona disassembly, while corona assembly is not overtly affected. This however only shows that the corona is less stable if this residue cannot be phosphorylated, not that its dephosphorylation is the crucial trigger of corona disassembly. Furthermore, the authors clearly indicate that the T422A mutation also prevents retention of CENP-E after corona disassembly, thus suggesting that phosphorylation of this residue may affect CENP-E beyond its mere participation in the corona. In conclusion, the authors' observations are compatible with the claim, but the authors do not prove it, nor do they detail the mechanism of corona disassembly, something on which I comment below. The three sentences should be modified accordingly.

We thank the reviewer for pointing out the textual elements that could be improved. We agree that these suggestions will help us to present our data more accurately. In the revised version of our manuscript, we have modified the three sentences according to the reviewers' comments.

-On page 9, the authors claim that ‘...the impact of phosphorylation on intra- or inter-molecular properties of CENP-E remains unknown’. This should be revised. Phosphorylation of CENP-E on residue 422 by Aurora kinases has been described previously (reference 19), and convincingly linked to the recruitment of PP1 phosphatase. This previous evidence contradicts the statement, because the previous work establishes an ‘inter-molecular’ property of CENP-E. In the current manuscript, lack of phosphorylation is shown to lead to a stable intramolecular interaction of the N- and C-terminal moieties of CENP-E, in turn proposed to inhibit the motor domain. The two models are not incompatible, because the intramolecular interaction demonstrated here may require dephosphorylation of additional phosphorylation sites, a hypothesis that the author do not discuss but that is in line with work by Espeut et al. (Mol Cell 2010), who demonstrated that CENP-E auto-inhibition can be relieved by MPS1 and CDK1 phosphorylation of the CENP-E C-terminal tail. A CENP-E intramolecular interaction was also discussed in Vitre et al. (MBoC 2014), where additional segments of the CENP-E kinetochore-targeting domain were shown to inhibit motor activity if positioned too closely to the motor domain.

Thus, I suggest that this sentence, and the following discussion, should be revised to give proper credit to the substantial existing evidence that CENP-E may exist in an auto-inhibited state. I also re-iterate that the present study advances, but does not unveil, the mechanism of corona disassembly, as this may be much more complex than implied by the study of a single phosphorylation site. Detailing the mechanism would require substantial additional work.

We are grateful to the reviewer for drawing our attention on an inaccurate statement regarding the earlier reported effects on inter-molecular properties of CENP-E, namely the binding of PP1 (Kim, Holland et al. Cell 2010). In the revised version of our manuscript, we corrected this omission by substantial textual changes, as well as by including a new set of data collected by analyzing two newly generated cell lines expressing the CENP-E point mutants covering the PP1-binding motif (new Supplementary Figure 5). We have also further elaborated on the two papers that we had previously cited in the context of CENP-E autoinhibition (Espeut et al. Mol Cell 2010; Vitre et al. MBoC 2014).

-Related to the comments above, also on page 9 the authors write ‘In the absence of T422 phosphorylation, the N-terminal domain co-immunoprecipitated stronger with the C-terminal domain, indicating that non-phosphorylated CENP-E has a higher predisposition to be in an auto-inhibited state’. In fact, what is being shown is an intra-molecular interaction, not an auto-inhibition, which is rather a hypothesis.

We agree with the reviewer and changed the sentence into: “...indicating that non-phosphorylated CENP-E has a higher predisposition for intra-molecular interaction between its N- and C-termini that may promote autoinhibition”.

-On page 10, the authors write ‘This suggests that the C-terminal fragment interacts and inhibits endogenous CENP-E, which is in line with a dominant negative effect caused by the expression of different C-terminal KT-targeting domain constructs of CENP-E’. Again, this seems to be the preferred argument, but there are plausible alternatives to it. The C-terminal region of CENP-E expressed by the authors to demonstrate the dominant-negative effect they describe is the kinetochore-targeting domain, which may also exercise its dominant-negative effects through displacement of endogenous CENP-E from kinetochores and from Dynein, and therefore not by inhibiting endogenous CENP-E. This possibility should be discussed in

the text.

In the revised manuscript, we discuss this possibility by adding: “An alternative explanation is that, because it contains the kinetochore-targeting domain, the C-terminal fragment induces its dominant-negative effects by displacing the endogenous CENP-E from kinetochores. However, the latter model could not explain polar accumulation of CENP-E”.

- 'Our phospho-antibody-based data suggest that CENP-E dephosphorylation happens in a gradual manner, starting from the region most distal from the centromeric AurB'. What is the evidence for this? The levels of P-T422 (Figure 2h) are expected to be maximal in nocodazole. What is the evidence that there is any dephosphorylated CENP-E at kinetochores under these conditions? Also, please note that the P-T422 signal seems to be internal even to CENP-C in these images, which is possible but counterintuitive. So, I would recommend caution in the interpretation of these images.

In the revised manuscript, we improved our data on the phospho-antibody analysis by: 1) using super-resolution SIM imaging, 2) quantifying much larger sample, and 3) presenting the standard deviations. The shift in peaks between CENP-E and pT422 antibody signals suggest the existence of a non-phosphorylated fraction. To remain careful in interpreting these data, we kept using “suggest” wording.

Minor points:

Abstract

- 'long-sought mechanism' should possibly read 'long-sought-for mechanism'

Corrected.

- There are various instances of 'This data...' – As 'data' is plural, the authors should write 'These data...'

Corrected.

- There is a bit of a disconnect in the manuscript between the first experiments demonstrating a role of Dynein in CENP-E 'stripping', described on page 5, and other experiments on the same topic discussed on page 7. These experiments are also described in the context of different figures, and I wonder if the narrative wouldn't benefit from coupling the description of these experiments.

Although we agree with the reviewer that the dynein-pointing data could be pooled, we prefer to keep the current narrative, presenting the descriptive results indicating microtubule- and dynein- based stripping in Figures 1 and 2, followed by the mechanism-based evidence presented in Figure 3.

- '...toroidal allocation...' – Please consider '...radial distribution...' as an alternative

Corrected.

- '...either of inhibitors...' Consider '...either inhibitor...'

Corrected.

- Throughout the manuscript, please consider removing adverbs like ‘intriguingly’, ‘strikingly’ and related adjectives. Their use is highly subjective and unnecessary.

We adjusted the writing style according to the reviewer’s suggestion.

Reviewer #3 (Remarks to the Author):

Kinetochores are large protein machines that drive cell division by forming complex, force-generating attachments to spindle microtubules. These functionalities are mediated by several kinetochore sub-domains, with the distal fibrous corona domain being the least well characterised. The fibrous corona is the most dynamic region of the kinetochore, expanding into a supramolecular meshwork before being disassembled, in part, by the dynein transport of corona cargoes along attached microtubules. Characterising corona disassembly is crucial for our understanding of how kinetochores permit accurate and timely cell division, an event that is essential for the growth, development and homeostasis of metazoans.

In this study, Eibes and colleagues have used engineered cell lines, live-cell imaging, siRNA-mediated depletion and molecular inhibition to show how an Aurora A-Aurora B-CENP-E axis controls chromosome congression and corona disassembly by dynein. Given the importance of corona disassembly for mitotic progression, and the scarcity of information on how it is regulated, this study represents an important advance of broad interest. Overall, the experiments are well performed and controlled, the conclusions are supported by the data and the manuscript is clearly written. Therefore, I support publication of this study in Nature Communications after the following revisions.

We thank the reviewer for recognizing the quality and importance of our study.

1. Can the authors clarify what data were used to calculate p-values? In some instances visually small differences have incredibly small p-values, which suggests to me that the number of kinetochores, rather than the number of independent experiments, have been used for the calculation. It would be helpful if the average for each biological replicate was added to the presented plots.

We thank the reviewer for this comment, and we agree that plotting the average kinetochore intensity of each cell is a better way of presenting our data than plotting each kinetochore value. We have modified the graphs accordingly (new Figures 1g, 2e and 2g) and clearly stated what has been plotted.

2. I am concerned about the accuracy of the images and associated quantification presented in figure 2h. In order to conclude that CENP-E and CENP-E pT422 are spatially separated I suggest the authors include at least two of the following. (1) Higher resolution imaging like the SIM data presented in Extended data figure 2b, (2) quantify a much larger sample and/or (3), plot distance measurements with error bars and associated statistics.

We are grateful to the reviewer for this important and constructive suggestion, which helped us to significantly improve the data presented in Figure 2h. Following all three recommendations, we have now included new SIM images coupled with quantifications of a

larger sample that is plotted with standard deviation.

3. Kinetochores are hard to distinguish in the images presented in Figure 4c. Can the authors include some zooms so that the staining patterns are clearer to the reader? Furthermore, can the authors comment on why the CENP-E WT signal shows a different spatial distribution between conditions. For example, CENP-E WT appears to localise further from the monopole in the spindly panel when compared to the p150 panel.

We agree with the reviewer that the p150 panel in Figure 4c can be misinterpreted and that the kinetochores in a close proximity to the spindle poles could be mistaken for spindle pole accumulation. To avoid any potential confusion, we have now modified the figure by including a more representative image (new Figure 4c).

The reason why kinetochores localize slightly closer to the spindle pole in some cells may be explained by the differences between total and partial RNAi rescue by GFP-CENP-E. In CENP-E-depleted monopoles, kinetochores are dragged towards the spindle pole due to a stronger activity of dynein (as shown in Barisic et al, Nat Cell Biol 2014, PMID: 25383660). To avoid potential effects of CENP-E overexpression, we depleted endogenous CENP-E using 3'UTR-targeting siRNAs, which can result in some cells not fully rescuing the RNAi phenotype.

REVIEWERS' COMMENTS

Reviewer #1 (Remarks to the Author):

The authors have addressed the reviewer's comments with additional experiments and further discussion. These changes addressed the concerns raised by the reviewer and improved the study.

Reviewer #2 (Remarks to the Author):

In this revision, the authors have addressed all main reviewers' concerns, both textually and with new experiments. On this basis, I strongly support publication of this nice study and wish to congratulate the authors.

Reviewer #3 (Remarks to the Author):

The authors have addressed all my concerns in the revised manuscript, which will be well received by the readership of Nature Communications.